# LDLR is used as a cell entry receptor by multiple alphaviruses

Xiaofeng Zhai[1,6], Xiaoling Li[1,6], Michael Veit [2,6], Ningning Wang[1], Yu Wang[1], Andres Merits[3], Zhiwen Jiang[1], Yan Qin[1], Xiaoguang Zhang[1], Kaili Qi[1], Houqi Jiao[1], Wan-Ting He[1], Ye Chen[4], Yang Mao [5] ✉ & Shuo Su[1] ✉

Alphaviruses are arboviruses transmitted by mosquitoes and are pathogenic to humans and livestock, causing a substantial public health burden. So far, several receptors have been identified for alphavirus entry; however, they cannot explain the broad host range and tissue tropism of certain alphaviruses, such as Getah virus (GETV), indicating the existence of additional receptors. Here we identify the evolutionarily conserved low-density lipoprotein receptor (LDLR) as a new cell entry factor for GETV, Semliki Forest virus (SFV), Ross River virus (RRV) and Bebaru virus (BEBV). Ectopic expression of LDLR facilitates cellular binding and internalization of GETV, which is mediated by the interaction between the E2-E1 spike of GETV and the ligand-binding domain (LBD) of LDLR. Antibodies against LBD block GETV infection in cultured cells. In addition, the GST-LBD fusion protein inhibits GETV infection both in vitro and in vivo. Notably, we identify the key amino acids in LDLR-LBD that played a crucial role in viral entry; specific mutations in the CR4 and CR5 domain of LDLR-LBD reduce viral entry to cells by more than 20-fold. These findings suggest that targeting the LDLR-LBD could be a potential strategy for the development of antivirals against multiple alphaviruses.

Alphaviruses (family *Togaviridae*) are a genus of the positive-strand RNA viruses. The majority of known alphaviruses are transmitted by mosquitoes to humans and other vertebrates, where they cause a range of clinical manifestations[1,2]. Old World alphaviruses, including Sindbis virus (SINV), chikungunya virus (CHIKV), Semliki Forest virus (SFV), Ross River virus (RRV), o'nyong-nyong virus (ONNV), Middelburg virus (MIDV), Mayaro virus (MAYV) and Getah virus (GETV), usually cause fever, maculopapular rash, arthralgia and myalgia, whereas New World alphaviruses, including Venezuelan equine encephalitis virus (VEEV), Western equine encephalitis virus (WEEV) and Eastern equine encephalitis virus (EEEV), can cause fatal encephalitis. Most of these viruses are endemic in tropical regions and occasionally cause large outbreaks[1–4].Despite their medical importance, no antiviral therapeutics are available to protect humans from alphavirus infection, which urged FDA to approve the first vaccine for CHIKV infection using the Accelerated Approval pathway recently[5,6].

The alphavirus virion is a spherical enveloped particle with a diameter of about 70 nm and a T = 4 icosahedral symmetry. The RNA genome of approximately 12 kb in length encodes four non-structural proteins (nsP1-4), which mediate virus genome replication, host subversion, and immune escape[7], and five structural proteins, including capsid (Cap), E3, E2, 6 K/transframe (TF) and E1[8,9]. The surface of virions contains 80 spikes, each of which are assembled by three heterodimers of E2-E1 glycoproteins[10–12]. The spike binds to receptors on

[1]Academy for Advanced Interdisciplinary Studies, Engineering Laboratory of Animal Immunity of Jiangsu Province, College of Veterinary Medicine, Nanjing Agricultural University, Nanjing, China. [2]Institute for Virology, Center for Infection Medicine, Veterinary Faculty, Free University Berlin, Berlin, Germany. [3]Institute of Bioengineering, University of Tartu, Nooruse Street 1, 50411 Tartu, Estonia. [4]Key Laboratory of Fujian-Taiwan Animal Pathogen Biology, College of Animal Sciences, Fujian Agriculture and Forestry University, Fuzhou 350002, China. [5]School of Pharmaceutical Sciences and National-Local Joint Engineering Laboratory of Druggability and New Drugs Evaluation, Sun Yat-sen University, Guangzhou, China. [6]These authors contributed equally: Xiaofeng Zhai, Xiaoling Li, Michael Veit. ✉e-mail: maoyang3@mail.sysu.edu.cn; shuosu@njau.edu.cn

the surface of the host cell. Bound virions are subsequently internalized principally by clathrin-mediated endocytosis. Acidification of the vesicle's interior triggers the fusion of the viral envelope with the vesicle membrane and the release of viral nucleocapsid and subsequently RNA genome into the cytoplasm[13].

Receptors with a clearly defined role in virus tropism and pathogenesis have been identified for some alphaviruses[14–16]. Matrix remodeling associated 8 (MXRA8) has been identified as a cellular entry receptor for GETV, CHIKV, MAYV, ONNV, RRV, SINV, and WEEV[14,17–19]. Low-density lipoprotein receptor class A domain containing 3 (LDLRAD3), highly conserved between vertebrates, is a receptor for VEEV[15]. In addition, very low-density lipoprotein receptor (VLDLR) from many species (including mosquitoes) and apolipoprotein E receptor 2 (ApoER2) have recently been identified as viral receptors in vertebrate and invertebrate cells for SFV, EEEV, and SINV[16]. In terms of mode of interaction, both LDLRAD3 and MXRA8 bind to the "canyon" between two protomers of the E spike on the surface of the alphaviruses, making simultaneous contact with E1 and E2. On the other hand, VLDLR binds to the DIII domain of the SFV E1 protein close to the envelope membrane and does not interact with the E2 of SFV[12,20,21]. However, it has been suggested that more than one receptor exists for CHIKV and RRV, for CHIKV titers remain high in tissues with MXRA8 receptor knocked-out and RRV is still capable of causing disease in the absence of MXRA8[22,23]. Furthermore, the expression of MXRA8 has not been reliably detected in several tissues that are infected by CHIKV, ONNV, RRV and MAYV[23]. Similarly, for SFV, ApoER2 is almost exclusively expressed in the central nervous system. Although VLDLR is expressed in multiple tissues, it has been observed that the absence of VLDLR or presence of antibodies against VLDLR do not completely block the infection of SFV in Vero, U2OS, A549, Huh7, and other cells[16,24]. Collectively, these evidences suggest the presence of additional receptors for alphaviruses. From the perspective of virus tropism, the ability to bind to more than one receptor may expand the repertoire of target tissues/organs of alphaviruses, lead to more severe and rapid development of symptoms, and also facilitate host or vector switching that may lead to virus outbreaks.

GETV, a once neglected and re-emerging mosquito-borne alphavirus, is currently posing a threat to many livestock and probably even humans. Notably, GETV infection has been reported in different mammalian orders other than their perceived reservoir host groups[25]. Therefore, although MXRA8 has been identified as a receptor for GETV, the rather limited expression profiles of MXRA8 in several tissues cannot explain the ability of GETV to infect large numbers of species and tissues[17,23,26]. It remained unknown whether receptors identified for other alphaviruses or some unknown receptors facilitate the cell entry of GETV[27].

Several known receptors for alphaviruses, such as VLDLR, LDLRAD3, and ApoER2, belong to the low-density lipoprotein receptor (LDLR) family[28]. Members of the LDLR family are remarkably conserved throughout evolution and share the same structural motifs: a ligand-binding domain (LBD) composed of varying numbers of cysteine-rich (CR) repeats, epidermal growth factor (EGF)-like domains, a transmembrane helix, and a cytoplasmic domain[29]. Aside from the involvement of VLDLR, ApoER2, and LDLRAD3 in the cell entry of certain alphaviruses, proteins of the LDLR family have also been shown to act as receptors for other viruses, including human rhinoviruses (HRVs), hepatitis C virus (HCV), vesicular stomatitis virus (VSV) and probably also dengue virus (DENV)[16,20,30–33]. These findings highlight the confounding nature of LDLR family as receptors for different viruses. It remains unclear whether these receptors are used differently in different cell types, or whether other proteins function as receptors for various alphaviruses[13,16].

Here, we present our serendipitous discovery of LDLR, the founding member of the LDLR family, as a receptor for various alphaviruses including GETV, SFV, RRV, and BEBV, and functionally mapped the interaction sites between LDLR and GETV glycoproteins. Pharmacological targeting of key contact residues in LBD of LDLR could lead to the development of novel strategies for the prevention and treatment of alphavirus infection.

## Results

### LDLR promotes infection of GETV and several other alphaviruses

To identify new alphavirus receptors, we first generated a reporter virus rGETV-mCherry with mCherry inserted immediately downstream of the furin cleavage site between E3 and E2 regions of the structural polyprotein (Supplementary Fig. 1a), so that the infection efficiency of rGETV-mCherry can be monitored by fluorescence microscopy analysis of mCherry expression (Supplementary Fig. 1b) and additionally verified by western blot analysis of the expression of E2-mCherry fusion protein and its precursor (Supplementary Fig. 1c). Compared to parental rGETV-HN, replication of rGETV-mCherry in BHK-21 cells was slightly reduced (Supplementary Fig. 1d) and the plaque size was also slightly smaller (Supplementary Fig. 1e).

To screen for proteins that can promote GETV infection, we transfected HEK 293T cells with a library of 150 randomly-selected membrane proteins (Supplementary Table 1, one individual plasmid was used per well), followed by infection with rGETV-mCherry. The transfection efficiency was measured by analysis of fluorescence of a co-expressed EGFP and plotted against mCherry fluorescence expressed by the reporter virus (Supplementary Fig. 2a). Surprisingly, even with a limited library of only 150 randomly-selected membrane proteins, we were able to identify two hits-LDLR and T cell immunoglobulin and mucin domain-containing protein 4 (TIMD4), both of which significantly promoted the infection of rGETV-mCherry as indicated by enhanced mCherry signal (Supplementary Fig. 2b).

To rule out a potential interference of mCherry on E2 to the results, we further generated another recombinant GETV reporter virus (rGETV-EGFP) for validation, which had EGFP expressed as an individual protein under the control of the viral subgenomic promoter (Supplementary Fig. 1). The promoted infection of rGETV-EGFP by LDLR and TIMD4 overexpression was further confirmed by qRT-PCR (Fig. 1a) and virus titers (Fig. 1b). However, the impact of overexpressing LDLR was at least 10-fold higher than that of TIMD4. At the same time, overexpression of LDLRAD3, a known receptor for VEEV (Supplementary Fig. 2c), did not increase rGETV-EGFP infection (Fig. 1a, b).

Next, we analyzed the effect of LDLR overexpression on the infection of other alphaviruses, including SFV, BEBV, RRV, MIDV, MAYV, VEEV, and CHIKV, as well as VSV, a rhabdovirus known to use LDLR as the receptor, using EGFP-expressing pseudo-viruses generated using a lentivirus vector (Supplementary Fig. 3a–c). In this assay we first prepared HEK 293T cell lines stably expressing hamster LDLR, LDLRAD3 or MXRA8 (Supplementary Fig. 4). Expression of these proteins was confirmed using flow cytometry, indirect immunofluorescence assay (IFA) and did not have a negative effect on cell viability (Supplementary Fig. 4a–c). It was found that expression of LDLR significantly increased the infection of HEK 293T cells with GETV, SFV, BEBV, RRV and VSV pseudo-viruses, while the infections with MIDV, MAYV, VEEV and CHIKV pseudo-viruses were not affected. In addition, we found that overexpression of MXRA8 significantly promoted the infection with pseudo-viruses of MAYV, CHIKV and RRV, which were previously shown to use this receptor[14], as well as with pseudo-viruses of BEBV, GETV and MIDV. It is worth noting that MXRA8 was considerably more effective than LDLR in promoting RRV infection. Overexpression of LDLRAD3 significantly promoted the infection with pseudo-virions of VEEV, which is consistent with previous reports (Fig. 1c–k)[14,15]. These results revealed that LDLR may be a receptor for multiple alphaviruses.

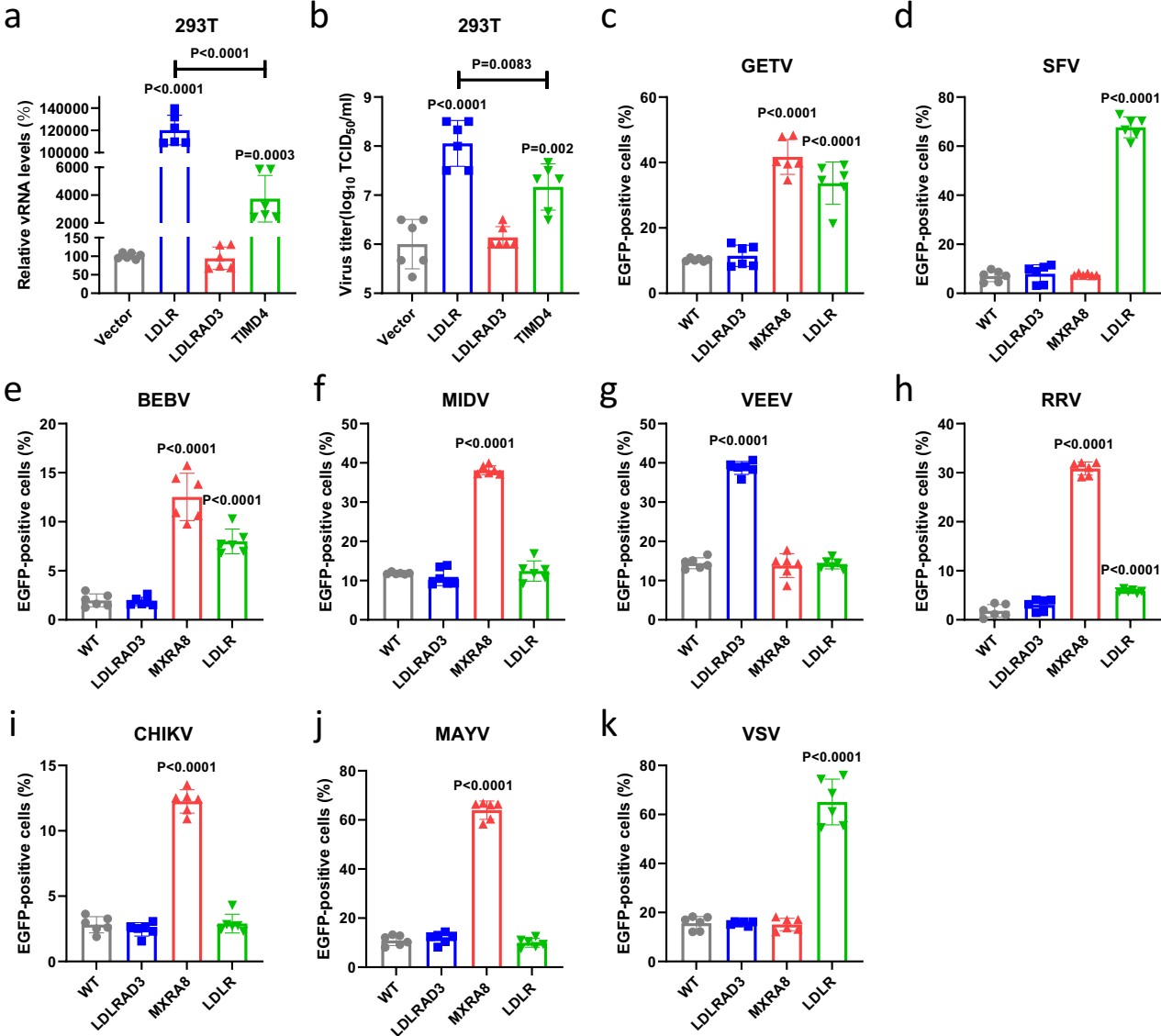

**Fig. 1 | Overexpression of LDLR significantly promotes the infection of various alphaviruses. a, b** HEK 293T cells expressing mouse LDLR, LDLRAD3 or TIMD4 or transfected with pCAGGS vector (negative control) were infected with the rGETV-EGFP. At 24 h post infection viral RNA (vRNA) levels in infected cells were measured via qRT-PCR (normalized relative to WT cells) (**a**) and viral titers in supernatant (**b**) were determined on BHK-21 cells. **c–k** HEK 293T cells stably overexpressing hamster LDLRAD3, MXRA8 or LDLR were infected with GETV (**c**), SFV (**d**), BEBV (**e**),

MIDV (**f**), VEEV (**g**), RRV (**h**), CHIKV (**i**), MAYV (**j**) or VSV (**k**) EGFP-expressing pseudo-viruses. EGFP-positive cells were counted using flow cytometry. Data are presented as mean values ± SD ($n$ = 3 independent experiments) and two-tailed $P$-values are calculated by unpaired Student's t test. Unless otherwise labeled, the displayed $P$-values are the significance between the experimental group and the control group (Vector or WT). Source data are provided as a Source Data file.

Lastly, to investigate whether the proviral effect of LDLR is host specie-specific, we also prepared HEK 293T cell lines stably expressing LDLR from pig (Supplementary Fig. 4) and used these to examine the effect of LDLR expression on the infection of different GETV strains. As shown in Supplementary Fig. 5, expression of either pig or hamster LDLR could promote infection of different GETV strains (GETV-HN, GETV-GX, GETV-FJ) as well as that of recombinant virus, rGETV-EGFP.

**LDLR increases viral infection by promoting the cell entry**
Next, we tested whether LDLR is directly involved in alphavirus entry using virus binding and internalization assays. As demonstrated in Fig. 2a, overexpressing LDLR significantly increased both the amount of bound authentic GETV at 4 °C and the amount of internalized GETV at 37 °C in HEK 293T cells. Using the recombinant reporter virus (rGETV-mCherry) and confocal microscopy imaging, we further

confirmed that overexpressing LDLR but not LDLRAD3 significantly promoted the adherence of rGETV-mCherry virions to the surface of HEK 293T cell at 4 °C and the internalization of rGETV-mCherry virions into cytoplasm of the cells at 37 °C (Fig. 2b). To find out whether LDLR is directly involved in cell entry of other alphaviruses, we generated and rescued recombinant SFV (rSFV-mCherry), RRV (rRRV-mCherry) and SINV-BEBV chimera (rSINV-BEBV-mCherry) (Supplementary Fig. 1). Wild type rSFV and rRRV were rescued as well; all these viruses were used to test their binding and internalization at 4 °C and 37 °C, respectively. Overexpression of LDLR significantly increased the binding of rSFV-mCherry, rSFV, rSINV-BEBV-mCherry, and, to a smaller extent, rRRV-mCherry and rRRV to HEK 293T cells (Supplementary Fig. 6); this data is in line with the results obtained using respective pseudo-viruses (Fig. 1d, e, h). These results suggested that LDLR was involved in the binding and internalization of virions of GETV and other alphaviruses.

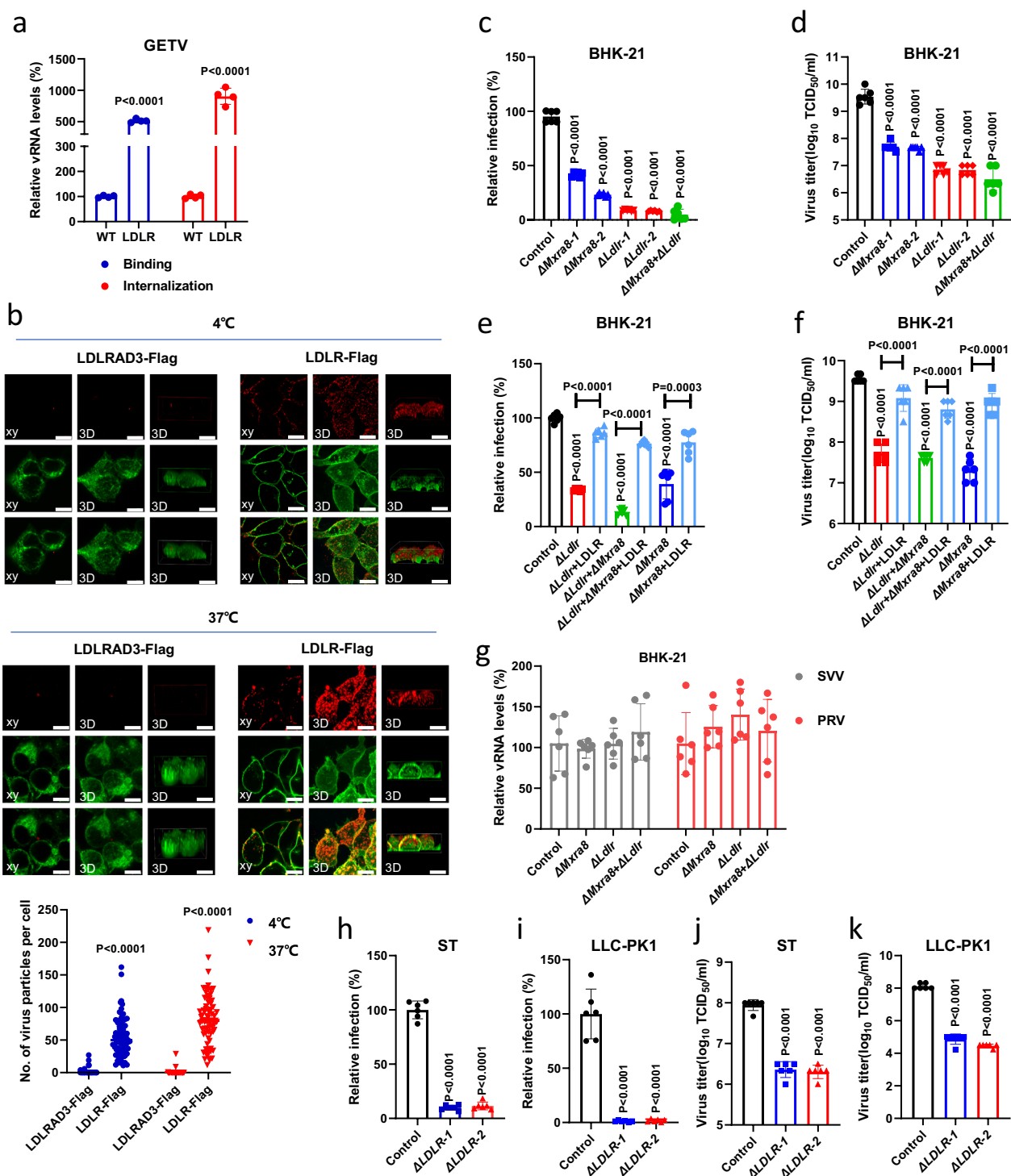

To further validate that LDLR and MXRA8 are indeed important host factors for GETV infection, we next used CRISPR-Cas9 to knock-out *Ldlr*, *Mxra8*, or both of them simultaneously in BHK-21 cells (Supplementary Fig. 7a–c). As demonstrated in Fig. 2c, d, deleting *Ldlr* and *Mxra8* alone or together significantly reduced the infection of rGETV-EGFP. Notably, the susceptibility of Δ*Ldlr* BHK-21 cells to infection of rGETV-EGFP was restored by transient over-expression of full-length LDLR. Furthermore, overexpression of LDLR also re-sensitized Δ*Mxra8* BHK-21 and Δ*Ldlr* + Δ*Mxra8* BHK-21 cells to infection of rGETV-EGFP (Fig. 2e, f, Supplementary Fig. 7). Conversely, overexpression of MXRA8 also restored virus infection in Δ*Ldlr* BHK-21 cells (Supplementary Fig. 7e, f). The ability of

MXRA8 and LDLR to compensate the loss of each other in the infection of rGETV-EGFP in BHK-21 is intriguing, because MXRA8 is very different in structure from all the members of the LDLR family. This suggested that GETV have multiple ways of interacting with host cells and the infection in vivo might be dependent on the actual expression of different receptors. In contrast, LDLR and/or MXRA8 deficiency did not affect the ability of non-enveloped Seneca Valley virus (SVV) or unrelated enveloped pseudorabies virus (PRV) to infect BHK-21 cells (Fig. 2g).

Because GETV has caused various outbreaks in pig farms in recent years[25], we selected two pig cell lines, common ST (porcine testicular cells) and LLC-PK1 (porcine kidney cells), to test the

**Fig. 2 | Expression of LDLR increases virus infection by promoting cell entry. a** Relative vRNA levels in WT HEK 293T cells and cells stably overexpressing hamster LDLR. For binding, cells were harvested 30 min after incubation with GETV-HN at 4 °C. For internalization, unbound viruses were removed and cells were incubated further at 37 °C for 1 h before harvest. vRNA levels were measured by qRT-PCR; data was normalized relative to WT cells. Data are presented as mean values ± SD (*n* = 4 independent experiments) and two-tailed *P*-values are calculated by unpaired Student's t test. **b** Orthogonal (left) and 3D (center, right) views of receptor and virus colocalization by confocal fluorescence imaging. Cells were transfected with LDLR-Flag or LDLRAD3-Flag plasmid and incubated with rGETV-mCherry virions before immunostaining with anti-Flag M2 antibody. Scale bar, 10 µm. The 3D image shows two different angles, from top to down (center) and from front to back (right). Data are presented as means ± SD (*n* = 75 cells examined over three independent experiments). **c, d** Flow cytometry analysis of WT, Δ*Mxra8*, Δ*Ldlr* or Δ *Mxra8*+Δ*Ldlr* BHK-21 cell lines infected with rGETV-EGFP (MOI = 0.001). Viral titers in supernatants were measured by titration on BHK-21 cells. **e, f** Flow cytometry analysis of Δ*Ldlr*, Δ *Mxra8*+Δ*Ldlr* or Δ*Mxra8* BHK-21 cells transfected with LDLR expression plasmid or mock-transfected prior rGETV-EGFP infection (MOI = 0.001). **g** qRT-PCR analysis of infection efficiency of Seneca Valley virus (SVV) and Pseudorabies virus (PRV) at MOI of 0.1 in WT, Δ*Mxra8*, Δ*Ldlr*, and Δ *Mxra8*+Δ*Ldlr* BHK-21 cells. Results are normalized to WT BHK-21 cells in (**c**–**g**). **h-k** Flow cytometry analysis of rGETV-EGFP (MOI = 0.1) infection in WT and Δ*LDLR* ST and LLC-PK1 cell lines. Results are normalized relative to WT cells. For flow cytometry, data are presented as mean values ± SD (*n* = 3 independent experiments) and two-tailed *P*-values are calculated by unpaired Student's t test (**c**–**k**). Unless otherwise labeled, the displayed *P*-values are the significance between the experimental group and the control group (WT, Control or LDLRAD3-Flag). Source data are provided as a Source Data file.

relevance of LDLR for GETV infection in pigs. Similar to results with BHK-21, rGETV-EGFP infection in Δ*LDLR* ST or Δ*LDLR* LLC-PK1 cell lines was also significantly reduced (Supplementary Fig. 7, Fig. 2h–k). Together, these results demonstrated an important role of LDLR in promoting the entry of GETV.

### The ligand-binding domain of LDLR binds to the E2-E1 spike of GETV

The extracellular part of LDLR consists of an N-terminal ligand-binding domain (LBD) with seven cysteine-rich (CR) repeats, a cluster of EGF-like modules containing a β-propeller domain, and a heavily *O*-glycosylated membrane-proximal domain[29]. To find out the region of LDLR involved in the entry of GETV, we first constructed truncated LDLRs with deletion of either the EGF-like or the LBD domain (Fig. 3a). These truncations did not affect the expression and membrane localization of LDLR (Supplementary Fig. 4b). As demonstrated in Fig. 3b–d, LDLR lacking the EGF-like modules still supported rGETV-EGFP infection in HEK 293T cells, but a mutant with the LBD domain removed failed to mediate the virus infection (Fig. 3b–d), which was further confirmed by western blot analysis of the expression of viral capsid (Cap) and E2 protein (Fig. 3e).

Next, we tested whether LBD of LDLR binds directly to GETV virions. For this purpose, we produced a GST-LBD fusion protein as described previously[30] (Supplementary Fig. 8a). Using a pull-down assay, we found that GST-LBD binds to GETV-HN virions, while GST does not. Likewise, a monoclonal antibody against E1 protein could pull down GST-LBD bound to GETV virions (Fig. 3f). Specifically, we confirmed the interaction of LBD with E proteins of GETV by pulling down a p62-E1 polyprotein with GST-LBD using either lysate of HEK 293T cells expressing p62-E1 (Fig. 3g) or purified GETV p62-E1-Strep protein (Supplementary Fig 8c, d, Supplementary Fig. 9e). Furthermore, we also examined the interaction between GST-LBD and p62-E1-HA of SFV, RRV, and BEBV. As shown in Supplementary Fig. 9a–d, GST-LBD can interact with p62-E1-HA polyproteins of all of these viruses.

The interaction between the LBD of LDLR and GETV particles was additionally confirmed using an enzyme-linked immunosorbent assay (ELISA). Here GETV virions were captured using a mouse anti-E1 monoclonal antibody (mAb). As demonstrated in Fig. 3h, GST-LBD but not GST was able to interact with bound virions.

Finally, we used the bio-layer interferometry analysis to measure the binding affinity between either purified GETV p62-E1-Strep or GETV virus-like particles (VLPs), which mimic the structure of native virions[34], and GST-LBD (Supplementary Fig. 8c–f). This analysis revealed that the GETV p62-E1-Strep and GETV VLPs specifically bound to GST-LBD but not to GST protein (Supplementary Fig. 9f, g). Taken together, the data above prove that there is direct binding between the LBD of LDLR and the E2-E1 spike of GETV.

### The GST-LBD fusion protein and anti-LDLR antibody block GETV infection in a dose-dependent manner

If GETV uses LDLR as a host cell receptor, the soluble LDLR-LBD should be able to inhibit GETV infection. To test this hypothesis, we performed in vitro neutralization assays using rGETV-EGFP virions and purified GST-LBD. We found that GST-LBD, but not GST, neutralized the infectivity of rGETV-EGFP in BHK-21 cells in a dose-dependent manner (Fig. 4a, b). Very similar results were obtained using LLC-PK1, Vero and ST cells (Fig. 4c, d). In addition, for further proof, we produced LBD-Fc fusion protein in Expi293F cells and repeated the in vitro neutralization assays. It was observed that both *E.coli* expressed GST-LBD and mammalian cell expressed LBD-Fc blocked virus infection in ST cells to a similar extent (Supplementary Fig. 10).

To confirm the biological significance of these findings, we next generated mouse polyclonal antibodies against LDLR (anti-LDLR) and tested whether these antibodies could block GETV infection. The purified anti-LDLR antibodies were validated and shown to react with LDLR but not with VLDLR (Supplementary Fig. 11, Supplementary Table 2). Pretreatment of BHK-21 cells with mouse anti-LDLR immune serum but not serum from naive mice reduced rGETV-EGFP infection in a dose-dependent manner (Fig. 4e, f). The inhibition was also observed for ST, LLC-PK1, and Vero cell cultures (Fig. 4g, h). Together, these results revealed that the interaction of the LBD domain of LDLR with GETV virions is required for GETV infection.

### Treatment with GST-LBD fusion protein reduces pathogenicity and virus titers in GETV-infected mice

To extend the analysis of the physiological role of the interaction between LDLR and GETV virions, we also evaluated whether treatment with GST-LBD could attenuate GETV infection in vivo. As demonstrated in Fig. 5a and b, subcutaneous administration of GST-LBD at 4.5 h after challenge with a lethal dose of GETV protected two-day-old mice against weight loss and significantly reduced mortality. Furthermore, compared to the GST control, treatment with GST-LBD significantly reduced virus titers in the spleen, ankle, and lung of two-day-old GETV-infected mice (Fig. 5c). Similarly, treatment with GST-LBD also reduced GETV titers in the spleen, ankle, and lung of six-week-old mice, as well as significantly reduced viremia, both at 24 h and at 48 h post infection (Fig. 5d).

### CR5 and CR4 regions of LDLR are essential for promoting GETV infection

The LBD of LDLR is made of 7 cysteine-rich repeats (CR1 to CR7, Fig. 6a)[29]. In order to narrow down the specific CR repeat mediating cell entry of GETV, we constructed a panel of LDLR mutants with each individual LDLR CR domain deleted. We next stably expressed these LDLR mutants in HEK 293T cells, which have similar steady state expression levels (Supplementary Fig. 4a) and membrane localization

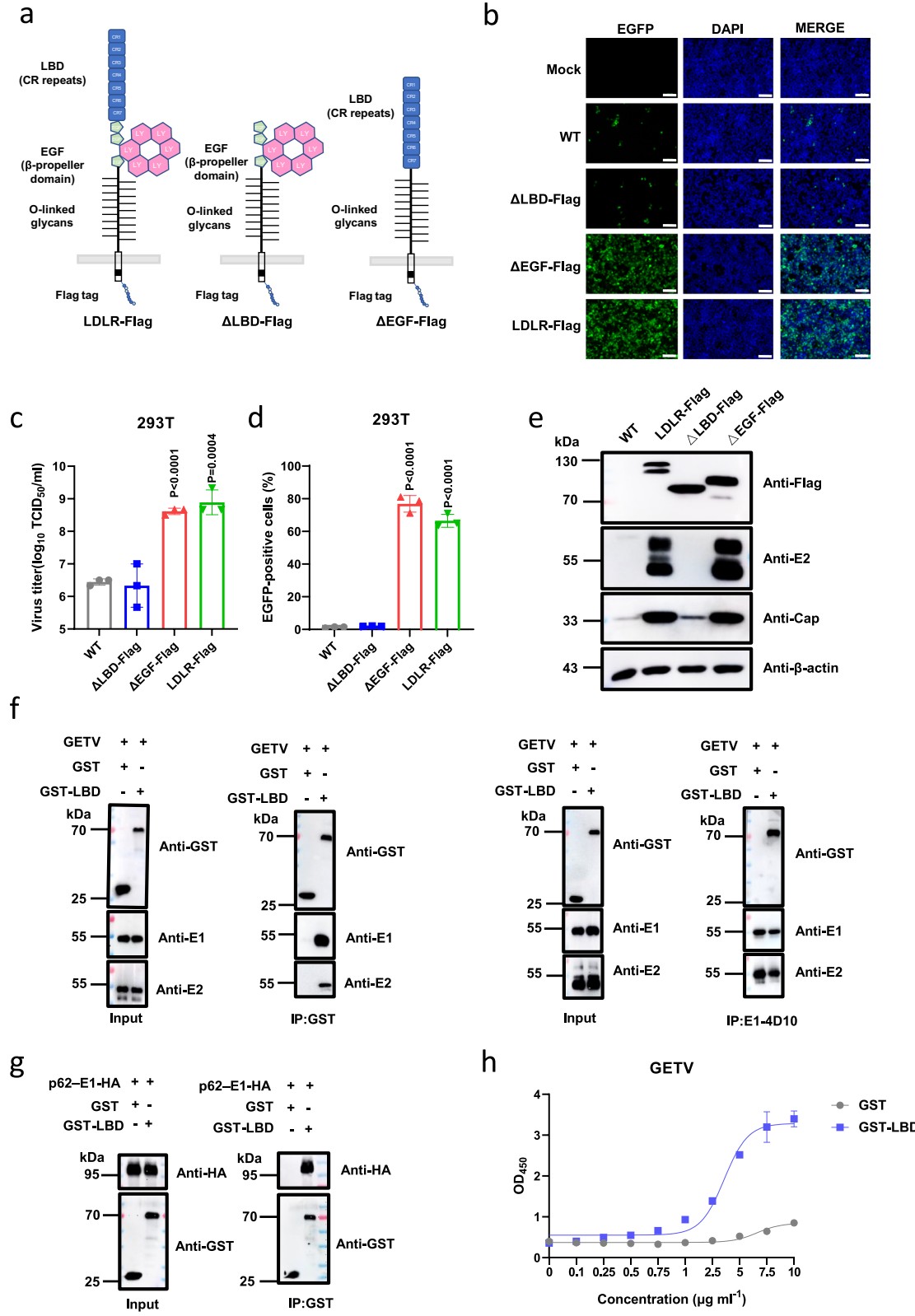

(Supplementary Fig. 4b). Infection of these cell lines with rGETV-EGFP revealed that deletion of CR5 completely eliminated the ability of LDLR to promote rGETV-EGFP infection. After deletion of CR4, numbers of EGFP positive cells and viral titers were still significantly higher compared with those in the group without overexpression of LDLR protein; however, the GETV infection was upregulated to a much lesser extent than in cells overexpressing full-length LDLR protein (Fig. 6b-d). This

data suggests that both CR4 and CR5 are essential for the infection of GETV, with CR5 playing the most dominant role.

Two binding sites were previously identified from the 3D-structure of VEEV particles bound to the D1 domain of LDLRAD3, another receptor of the LDL family. One was called "wrapped", and in this site D1 contacts both with E2 and E1. Another site was called "intraspike", and interacts only with E2 (Fig. 6e). Superposing the

**Fig. 3 | LBD of LDLR directly binds to GETV E2/E1 proteins. a** Schematic presentation of used LDLR constructs showing ligand-binding domain (LBD) consisting of 7 CR repeats, 3 EGF-like modules with β-propeller domain, O-linked glycans domain, transmembrane region and a cytoplasmic tail. **b–d** WT HEK 293T cells, HEK 293T stably expressing Flag-tagged LDLR from pig or its deletion mutants (ΔLBD and ΔEGF) were infected with rGETV-EGFP at an MOI of 1. At 18 h post infection cells were imaged using fluorescence microscopy. Scale bar, 500 μm (**b**). Viral titers in supernatants were determined on BHK-21 cells (**c**) and EGFP-positive cells were counted using flow cytometry (**d**). Data are presented as mean values ± SD (n = 3 independent experiments) and two-tailed P-values are calculated by unpaired Student's t test. **e** Lysates of infected cells were analyzed by western blot using antibodies against the Flag-tag, E2 and capsid (Cap) proteins of GETV, β-actin was used as loading control. **f** Co-immunoprecipitation of GETV virions and GST-LBD

fusion protein. Virions were incubated with GST or GST-LBD and subsequently immunoprecipitated with antibodies against GST (left part) or E1 (right part). Samples were subjected to western blot analysis using antibodies against GST or viral E1 or E2 proteins. Input: 10% of samples before immunoprecipitation. **g** HA-tagged p62-E1 of GETV was incubated with soluble GST or GST-LBD proteins and subsequently subjected to immunoprecipitation with anti-GST antibody. Proteins were detected using western blot analysis and antibodies against GST and HA tag. **h** Binding of GETV-HN virions to soluble GST and GST-LBD proteins detected using ELISA. Data are presented as mean values ± SD (n = 3 independent experiments) and two-tailed P-values are calculated by unpaired Student's t test. $OD_{450}$, optical density at 450 nm. The experiment was performed twice with similar results. Molecular masses are indicated in kDa (**e–g**). Source data are provided as a Source Data file.

structures of CR5 of LDLR and D1 of LDLRAD3 revealed a high degree of structural homology. To identify specific residues in CR5 of LDLR that may bind to E2 of GETV, we selected seven residues located at positions equivalent to the amino acid residues in D1 of LDLRAD3 known to be involved in binding to VEEV virions. In D1 of LDLRAD3 three of these residues contribute to the intraspike form, three to the wrapped form, and one (W47) is involved in binding to both forms (Fig. 6f). We generated a panel of eleven mutant LDLRs. Seven single mutations (H11T, E16R, S20R, W22I, D25K, G26H, D32K) and their combination (11T-32K) were introduced into CR5. Three single substitutions were also introduced into CR4 (W19I, D22K, D29K). In the full-length LDLR protein these substitutions are W165I, D168K, D175K, H205T, E210R, S214R, W216I, D219K, G220H, D226K, and 205T-226K. We next stably expressed these LDLR mutants in HEK 293T cell lines. Again, it was observed that mutant proteins had approximately the same steady-state expression levels (Supplementary Fig. 4a) and had maintained their membrane localization (Supplementary Fig. 4b). These cell lines were then infected with rGETV-EGFP and the efficiency of infection was analyzed using flow cytometry and titration as described above. As demonstrated in Fig. 6g and h, mutations at positions 165, 168, 175, 210, 216, 219, and 226 all reduced LDLR-mediated virus infection. The D226K substitution had the most significant impact, reducing the titer of rGETV-EGFP nearly as much as did the combination of all seven substitutions in the CR5.

Finally, since CR5 of LDLR and D1 of LDLRAD3 share a similar fold, we asked whether chimeric LDLRAD3, where the native D1 domain is replaced by the CR5 domain of LDLR, can serve as a GETV receptor. Similarly, we investigated if the reciprocal swap would enable VEEV to use chimeric LDLR as a receptor. It was found that the chimeric receptors were expressed at a similar level as their unmodified counterparts (Supplementary Fig. 12). Nevertheless, the LDLRAD3 harboring CR5 domain of LDLR could not promote GETV pseudo-virus infection, neither could LDLR harboring D1 domain of LDLRAD3 promote VEEV pseudo-virus infection.

## Discussion

The affinity-based interaction between alphavirus particles and cellular receptors is a crucial determinant for virus host range, tissue tropism and pathogenesis. In this work, we identified LDLR as a novel receptor used by several alphaviruses that belong to the Semliki Forest antigenic complex: GETV, SFV, BEBV and RRV. We also show that other alphaviruses, such as VEEV, CHIKV, MAYV and MIDV cannot use LDLR for cell entry. VEEV, but not any of the other alphavirus tested here, uses LDLRAD3 as a receptor. We also confirmed that all alphaviruses tested, except SFV and VEEV, can use MXRA8 as one of the cell entry factors (Fig. 1). However, our knockout results also revealed that LDLR and MXRA8 are not the only receptors for GETV in BHK-21 cells (Fig. 2d and f). Because SFV uses VLDLR and ApoER2 as receptors[16], which were not on the list of our limited library of membrane proteins yet also belong to LDLR family, we tested whether these two receptors could mediate cell entry of GETV. Not surprisingly, both VLDLR and

ApoER2 significantly promoted the infectivity of authentic GETV in BHK-21 cells (Supplementary Fig. 13). The association of the infectivity of certain alphavirus with LDLR family proteins is intriguing but requires more systematic analysis in the future to confirm this as a general rule.

Nevertheless, the ability to use multiple receptors is thus common feature for alphaviruses (Supplementary Table 3). Whether the receptors identified for GETV must both be present in a cell to make it fully susceptible or whether they can be used alternatively is also an important area worth future investigation. One could imagine that the first receptor acts as an attachment factor that accumulates virions on cell surface and the second receptor then enables virus particles to enter into the cytoplasm. Endocytosis of the LDLR has been very well documented[29]. We hypothesize that the E2-E1 protein spikes anchor the virion of GETV to LDLR receptors on cell surface, which then mediate virus entry by endocytosis. The subsequent acidification of the endosome leads to dissociation of LDLR and E2-E1 protein complexes, which further promotes the conformational change of E1 required for membrane fusion and releases of viral nucleocapsids to the cytoplasm.

We also demonstrate that the promotion of GETV infection depends on the CR4 and CR5 domains of LDLR, and identify the key amino acid residues involved in the entry of GETV virions. The amino acids in D1 of LDLRAD3 known to contact E2-E1 protein complexes of VEEV were used to select seven equivalent amino acids in CR5 of LDLR as elements of putative binding sites to E2-E1 of GETV (Fig. 6e, f). Substitution of most of these residues indeed resulted in the inability of the LDLR to promote cell entry of GETV. The only exception was the H11T substitution in CR5 of LDLR (position 205 in the full length of LDLR), which had no effect on cell entry of GETV. This contrast to the effect reported for the corresponding M36T substitution in LDLRAD3 that completely blocks cell entry of VEEV. Two mutations in CR5 of LDLR, S20R (residue 214 of LDLR), and G26H (residue 220 of LDLR) had only a moderate effect on the virus entry. The amino acid substitutions in CR5 of LDLR that had the strongest impact on GETV entry were W22I (residue 216 of LDLR), D25K (residue 219 of LDLR), and D32K (residue 226 of LDLR). Interestingly, substitutions in corresponding positions of the CR4 domain of LDLR also reduced GETV infection, indicating that GETV can bind to both CR4 and CR5. In contrast, the D2 domain of LDLRAD3 does not contribute to VEEV binding. The D1 domain of LDLRAD3 and CR4 and CR5 of LDLR have the same structure fold; nevertheless, GETV can only use LDLR but not LDLRAD3, whereas the opposite is true for VEEV (Fig. 1). This likely resulted from surface representations of these receptors (Supplementary Fig. 14) as many sites in these proteins that are important for E2-E1 binding are occupied by different amino acid residues. Likewise, the electrostatic surface potentials of these receptor molecules are also slightly different. Apart from the strongly acidic region, where the four acidic residues coordinate the calcium ion, the rest of the structure in CR5 is slightly acidic, whereas in D1 it is slightly basic (Supplementary Fig. 14). We also swapped the CR5 and D1 domains between LDLR and LDLRAD3, but

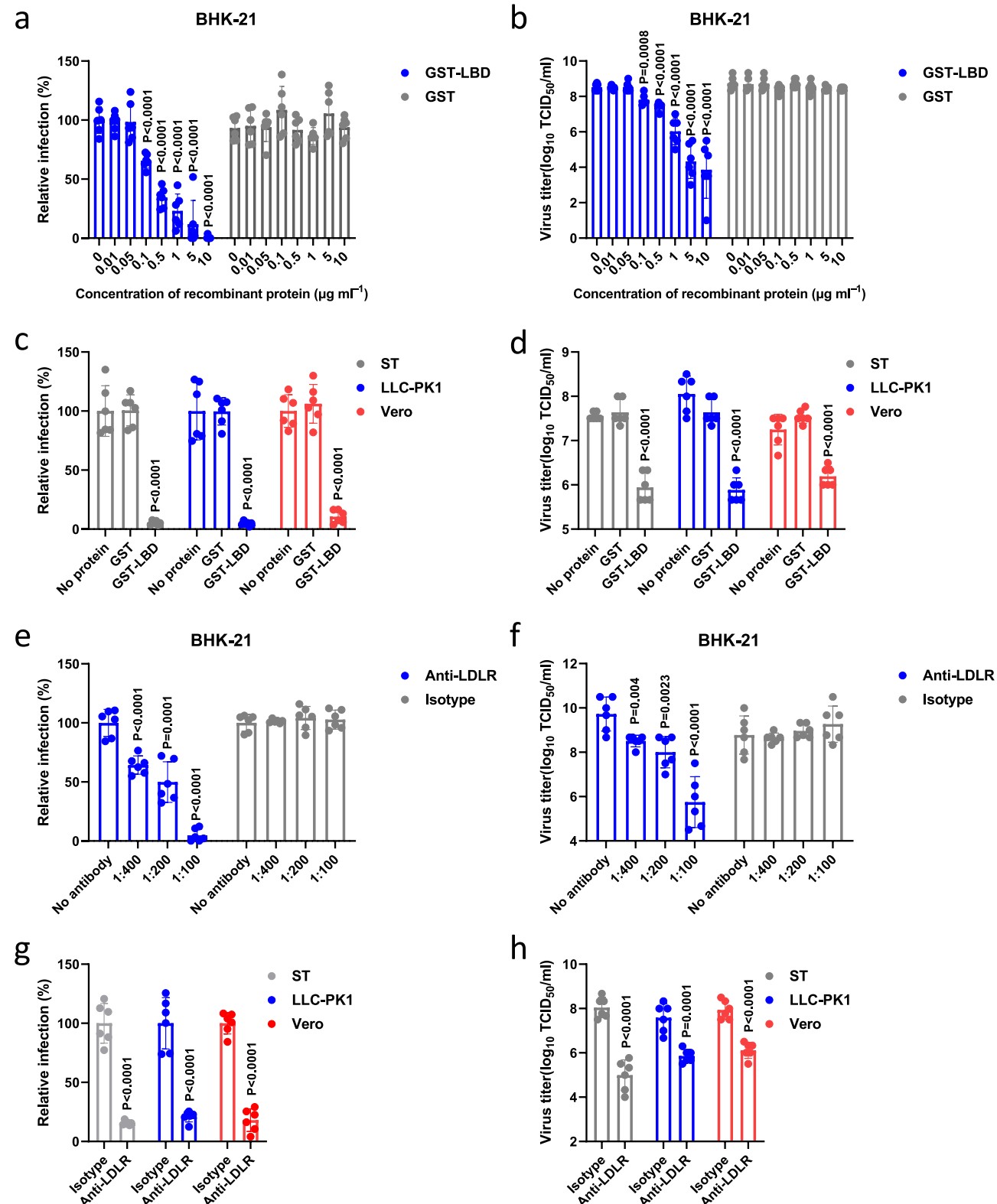

the chimeric receptors did not confer cell entry of GETV or VEEV. Thus, other factors, such as accessibility for the E2-E1 proteins in the spike, may play a role in these proteins acting as cell entry factors for specific viruses; it may also be important that LDLRAD3 is much smaller than LDLR[29].

Sequence alignment of CR4 or CR5 of LDLR proteins from human, mouse and livestock known to be hosts for GETV (pig, horse, cattle, goat, rabbit) show that the key amino acids for virus

entry are completely conserved. Comparison vertebrate proteins with their mosquito homolog also revealed high degree of similarity. This might be part of the reason why a number of alphaviruses are evolved to use LDLR as cell entry factor – using a conserved receptor allows infection of many host species. In addition, we also aligned the 3D-structure of CR5 with these of two other CR domains structures of which are known (CR2 and CR6). It was found that two to three of the key amino acid residues (based on our data from

**Fig. 4 | GST-LBD fusion protein or antisera against GST-LBD inhibit GETV infection in different cell lines. a**, **b** Virions of rGETV-EGFP were pre-incubated with increasing concentrations of GST-LBD or GST and then used to infect BHK-21 cells. At 18 h post infection number of infected cells was measured by flow cytometry (**a**) and virus titers in supernatant were determined using titration on BHK-21 cells (**b**). For panel **a** values obtained for cells infected with mock-treated virions were taken as 100%. Data are presented as mean values ± SD (n = 3 independent experiments) and two-tailed P-values are calculated by unpaired Student's t test. **c**, **d** Virions of rGETV-EGFP were pre-incubated with GST-LBD or GST proteins taken at concentration 10 μg ml⁻¹ or mock-treated and used to infect ST, LLC-PK1 or Vero cells. At 18 h post infection number of infected cells was measured by flow cytometry (**c**) and virus titer were determined using titration on BHK-21 cells (**d**). For

panel **c** values obtained for cells infected with mock-treated virions were taken as 100%. Data are presented as mean values ± SD (n = 3 independent experiments) and two-tailed P-values are calculated by unpaired Student's t test. **e**–**h** Cultures of BHK-21 (**e** and **f**), ST, LLC-PK1 or Vero cells (**g** and **h**) were preincubated with serial dilutions of anti-LDLR polyclonal mouse serum or an isotype control (naïve mice serum) before infection with rGETV-EGFP at an MOI of 0.001 for BHK-21 or 0.1 for ST, LLC-PK1 and Vero cells. At 18 h post infection the number of infected cells was measured by flow cytometry (**e** and **g**), and viral titer were determined using titration on BHK-21 cells (**f** and **h**). For panels **e** and **g** values obtained for control cells were taken as 100%. Data are presented as mean values ± SD (n = 3 independent experiments) and two-tailed P-values are calculated by unpaired Student's t test. Source data are provided as a Source Data file.

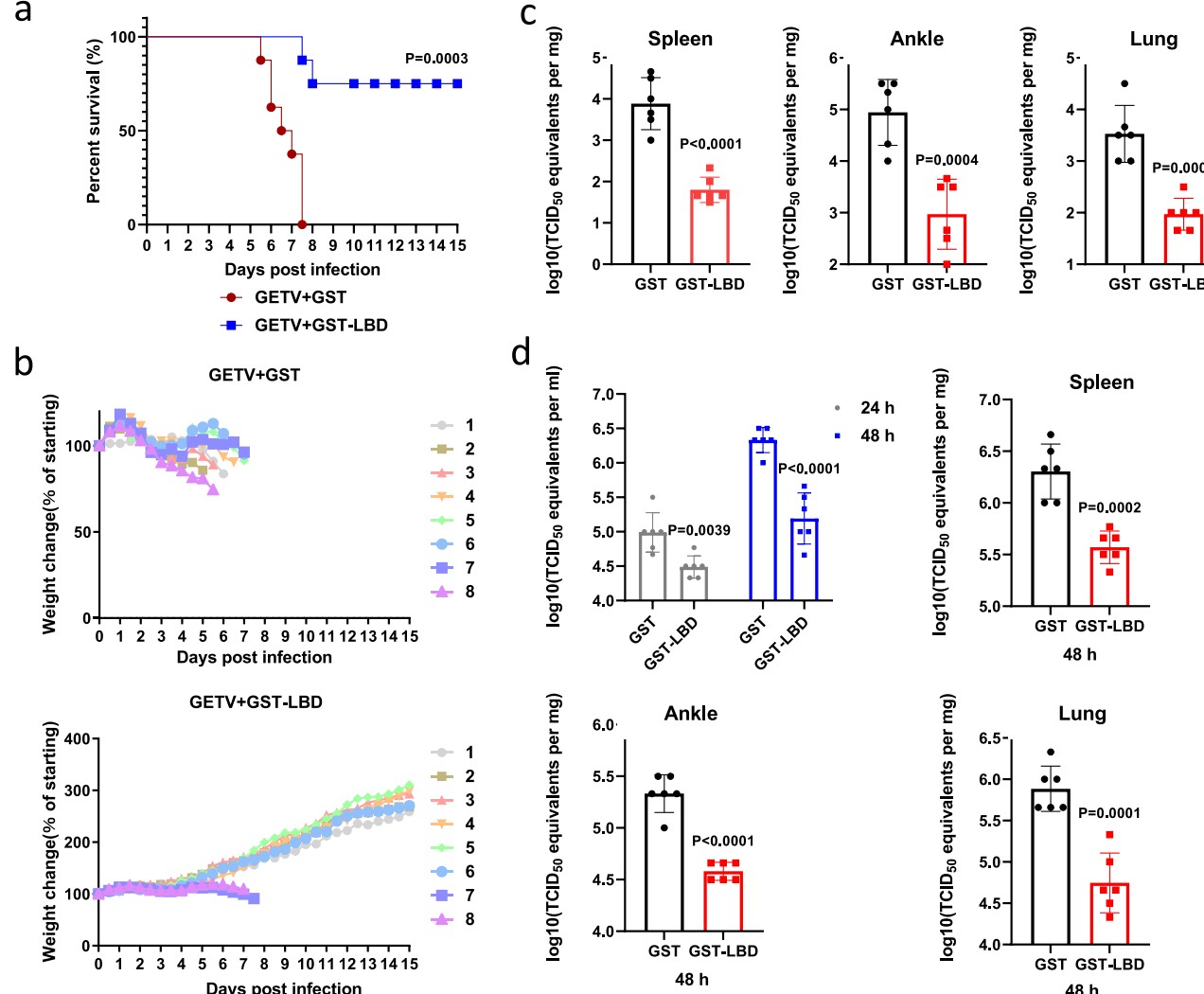

**Fig. 5 | Treatment with GST-LBD protein protects mice against GETV infection. a**–**c** Two-day-old ICR mice were infected with GETV and 4.5 h later subcutaneously inoculated with 100 μg GST-LBD or GST fusion protein (n = 8 for both treatment groups). Kaplan-Meier blot of survival data. Comparison of survival curves was performed by log-rank test, P = 0.0003 (**a**). Mice were monitored for weight change (**b**). Viral titers in tissues of another group of two-day-old mice infected and treated as described for **5a** were assessed 6 days post-infection (n = 6 for both treatment groups) (**c**). **d** Six-week-old ICR mice were subcutaneously infected with GETV; 5 h

later mice were intraperitoneally inoculated with 700 μg GST-LBD or GST (n = 6 for both treatment groups). Viral titer in serum was measured at 24 or 48 h post-infection, the viral titers in tissues were measured at 48 h post infection. All titrations were made on BHK-21 cells. Mean ± SD are shown, dots designate individual animals. Two-tailed P-values are calculated by unpaired Student's t test (**c**, **d**). Source data are provided as a Source Data file.

mutagenesis of CR5) are not present in CR2/CR6. This finding might partially explain why CR2 and CR6 are dispensable for the cell entry of GETV (Supplementary Fig. 15).

In conclusion, our study discovered LDLR as an important receptor for the cell entry of multiple alphaviruses, helping to

explain why alphaviruses, such as GETV, have a broad host range and tissue tropism. High-resolution structural information similar to that recently obtained for SFV and its receptor VLDLR[20] would be needed to define the complete interaction interface between LDLR and glycoproteins of alphaviruses. Nevertheless, because

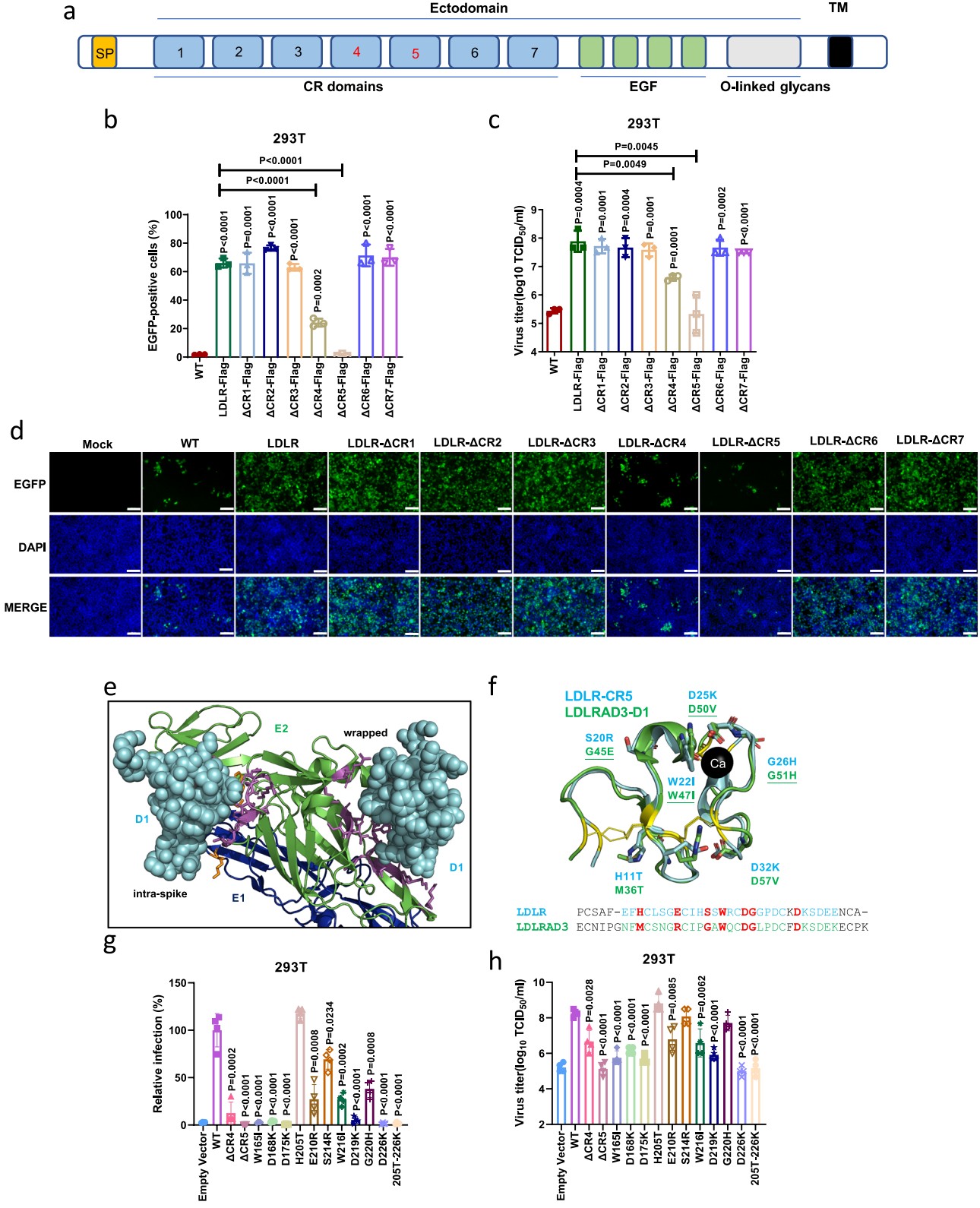

**Cells and viruses**

HEK 293T (ATCC CRL-3216), BHK-21 (ATCC CCL-10), LLC-PK1 (ATCC CL-101), ST (ATCC CRL-1746) and Vero cells (ATCC CCL-81) were cultured at 37 °C and 5% $CO_2$ in Dulbecco's Modified Eagle Medium (DMEM, Biological Industries, C3113-0500) supplemented with 10% fetal bovine serum (FBS, Biological Industries, C04001500). Expi293F cells (Thermo Fisher, A14635) were cultured in suspension at 37 °C and 8% $CO_2$ in Expi293 Expression Medium.

recombinant GST-LBD protein was shown to protect mice against GETV infection, LDLR could serve as a future target for drug development to treat alphavirus infection.

## Methods

The animal experiments were approved by the Institutional Animal Care and Use Committee of Nanjing Agricultural University, Nanjing, China (permission SYXK2017-0007; February 2017).

**Fig. 6 | Identification of domains and amino acid residues important for the functional interaction between LDLR and GETV. a** Scheme of the LDLR protein showing the N-terminal LBD composed of seven CR repeats, a cluster of EGF modules containing a β-propeller domain, and a membrane-proximal O-linked glycans domain. SP: signal peptide; TM: transmembrane region. **b–d** HEK 293T cells stably expressing pig LDLR or mutants lacking one of CR repeats were infected with rGETV-EGFP (MOI = 1, 12 h). EGFP-positive cells were counted by flow cytometry (**b**) or visualized by fluorescence microscopy (**d**), and viral titers in supernatants were measured by titration on BHK-21 cells (**c**). Data are mean ± SD of three biological replicates. Scale bar, 500 μm. **e** N-terminal part of the E2/ E1 heterodimer of VEEV shown as cartoon (E2 in green, E1 in blue) bound to D1 domain of LDLRAD3 shown as cyan spheres. D1 binds to two different sites, named intraspike and wrapped, respectively. Amino acids in D1 are shown as sticks, magenta for E2 and orange for E1. **f** Structural alignment of CR5 of LDLR (cyan, PDB: 1AJJ) and D1 of LDLRAD3 (green, PDB: 7N1H, RMSD for alignment = 0.631). The six cysteines forming three disulphide-bonds are colored in yellow. Seven amino acids invoved in D1 binding to E2 of VEEV are shown as green sticks and are underlined if they contribute to the intraspike form. The corresponding residues in CR5 of LDLR are shown as cyan sticks. Ca: calcium ion. The lower part: sequence alignment of D1 and CR5: seven amino acids chosen for mutagenesis in LDLR and the corresponding residues in LDLRAD3 are colored in red. **g, h** Flow cytometry analysis of rGETV-EGFP (MOI = 1, 12 h) infection in HEK 293T cells stably expressing LDLR or mutant LDLR. Results were normalized to cells expressing wt LDLR. Viral titers were measured on BHK-21 cells. Data are mean ± SD of four biologically replicates. Two-tailed P-values are calculated by unpaired Student's t test, the displayed P-values are the significance between the experimental group and the control group (Control or WT) (**b, c, g,h**). Source data are provided as a Source Data file.

Three GETV strains, named GETV-GX (MZ736796), GETV-HN (MZ736801), and GETV-FJ (MZ736799) were previously isolated from pigs. Construction of recombinant viruses designated rGETV-HN and rGETV-EGFP has been reported previously[35]. To make an infectious cDNA (icDNA) clone for rGETV-mCherry the sequence coding for mCherry reporter was inserted between regions encoding for E3 and E2 glycoproteins (downstream of the furin cleavage site) of GETV. The icDNA clones of rSFV-mCherry, rRRV-mCherry were constructed using corresponding icDNA plasmids[36,37] and had similar design; in a icDNA of rSINV-BEBV-mCherry, a region corresponding to structural proteins in SINV icDNA[38] was replaced with that of BEBV and sequence encoding for mCherry was inserted as described above. The strains of PRV and SVV were from a collection stored in our laboratory. All recombinant viruses, including wild-type rSFV and rRRV, were rescued, propagated, and titrated on BHK-21 cells; for titration, the endpoint method (determination of TCID$_{50}$) was used.

### Screening of viral entry factors through the use of host membrane protein expression library
A library containing plasmids for overexpression of 150 murine membrane proteins fused to Flag tag in the C terminus of target protein was constructed in-house. All plasmids in this library were designed to express EGFP reporter, which is expressed as a separate protein. The screening process was performed as follows. The 96-well cell culture plate was coated with L-Lysine homopolymer hydrobromide for 10 min at 37 °C. Then 0.2 μg of each plasmid and 0.4 μl of ExFect Transfection Reagent (Vazyme, T101) were diluted in 20 μl Opti-MEM, incubated for 5 min, then mixed and incubated for 15 min. The mixture was added to the 96-well cell culture plate together with trypsinized HEK 293T cells ($2 \times 10^4$ cells per well). 24 h later cells were infected with rGETV-mCherry (MOI of 1). The efficiency of infection with rGETV-mCherry (red fluorescence) and the transfection efficiency with plasmids expressing membrane proteins (green fluorescence) were visually observed using inverted fluorescence microscope; the fluorescence intensity was measured with an automatic microplate reader (TECAN Infinite M200 Pro).

### Production of pseudotyped virus particles
The sequences encoding for glycoproteins of GETV strain HeN202009-2 (GenBank: MZ736801.1), SFV strain SFV4 (GenBank: AKC01668.1), BEBV (GenBank: YP_005351239.1), VEEV Trinidad donkey strain (GenBank: AAC19322.1), RRV strain K70883 (GenBank: QFR08126.1), CHIKV strain LR2006-OPY1 (GenBank: ABD95938.1), MAYV strain BeH407 (GenBank: QDL88200.1) or MIDV strain Ar-749 (GenBank: UIX56005.1) were codon-optimized for expression in human cells, obtained as synthetic DNAs and used to substitute the G gene of VSV in the lentivirus packaging vector pMD2.0 G (gift from Didier Trono, Addgene plasmid #12259). To produce pseudotyped lentivirus particles, HEK 293T cells grown in 6-well cell culture plate were co-transfected with 0.5 μg obtained plasmids or pMD2.0 G, 1.5 μg

psPAX2 (gift from Didier Trono, Addgene plasmid #12260) and 2 μg retrovirus plasmid pshRNA, which also contains gene for EGFP[39]. The supernatant containing released particles was harvested 3 days post transfection, filtered using a 0.22 μm filter (Millipore, SLGP033N), aliquoted and stored at -80 °C until use.

### Plasmid construction and ectopic expression of receptor proteins
The cDNAs of *Mesocricetus auratus* LDLR, LDLRAD3, MXRA8, VLDLR, ApoER2 or *Sus scrofa* LDLR mRNAs were synthesized using total RNA isolated from BHK-21 or ST cells. To generate the sequences for expression of receptor proteins with C-terminal Flag-tag, their coding sequences were PCR amplified with the designed specific pairs of primers (Supplementary Table 4), and then cloned into the pCAGGS vector (NovoPro #V008798) for transient transfection or pCDH-CMV-MCS-EF1-Puro vector (SBI Systems Biosciences #CD510B-1) for packaging into lentivirus particles.

To obtain lentivirus particles, HEK 293T cells grown in 6-well cell culture plate were co-transfected using the 2 μg objective plasmids, 1.5 μg psPAX2 and 0.5 μg pMD2.0 G using ExFect Transfection Reagent (Vazyme, T101). Supernatants containing lentivirus particles were collected at 48 h post transfection and used to infect the target cells. After 48 h, puromycin was added. Individual puromycin-resistant colonies were picked 7 days later and expanded to obtain a pure cell line. The expression of intended proteins was verified by flow cytometry or western blot.

### Flow cytometry
At the selected time points cells expressing marker with green fluorescence were trypsinized, resuspended in PBS, and quantified by flow cytometry (BD Accuri® C6). For staining of cells that express Flag-tagged proteins or for counting of cells infected with wild type GETV-HN, GETV-GX, and GETV-FJ, adhered cells were fixed with 4% (w/v) paraformaldehyde (PFA) for 30 min, and then permeabilized with 0.1% Triton X-100 in PBS for 20 min. Cells were incubated with appropriate specific antibody as indicated in the figure legends for 2 h at room temperature. After washing three times with 0.01% Triton X-100 in PBS, cells were incubated with Alexa Fluor 488 conjugated goat anti-mouse IgG diluted in 0.01% Triton X-100 in PBS for 30 min at room temperature and analyzed by flow cytometry (BD Accuri® C6).

### Infection assay
For the infection with GETV-HN, GETV-GX, GETV-FJ, rGETV-EGFP, rGETV-mCherry (MOI of 0.001 for BHK-21 cells, 1 for HEK 293T cells, 0.1 for LLC-PK1, Vero and ST cells), cells were incubated with virus for 1 h at 37 °C, then washed three times with PBS, and supplemented with DMEM containing 2% FBS and incubated for 12, 18 or 24 h. To quantify the cells infected with rGETV-EGFP, the cells were trypsinized, collected in PBS, and the EGFP-positive cells were counted by flow cytometry (BD Accuri® C6), and the obtained data was analyzed using

FlowJo software. To quantify the infection with GETV-HN, GETV-GX or GETV-FJ, the infected cells were first stained for GETV E2 protein using 8D5 anti-E2 monoclonal antibody (in-house) and the relative infection was measured by flow cytometry as described above.

To assess the release of GETV virions, supernatants of infected cells were collected and viral titer measured on BHK-21 cells. In short, BHK-21 cells at a 96-well cell culture plate were incubated with ten-fold serial dilutions of supernatants for 48 h and the viral titers in TCID$_{50}$ units were calculated using Reed-Muench method.

For pseudo-virus experiments, particles harboring glycoproteins of GETV, RRV, BEBV, SFV, MAYV, MIDV, CHIKV, or VEEV were used to infect HEK 293T cells overexpressing mLDLR, mLDLRAD3 or mMXRA8 at MOI of 1. Infection efficiency was assessed 36 h later by counting EGFP-positive cells using flow cytometry.

For the infection of BHK-21 cells with PRV and SVV, cells were incubated with virus (MOI of 0.1) for 1 h at 37 °C, then washed three times with PBS, supplemented with DMEM containing 2% FBS and incubated for 18 h. The cells were lysed in TRizol Up reagent (TransGen Biotech, ET111-01-V2) and RNA extraction was performed using manufacturer's protocol. The qRT-PCR was performed using specific primers (Supplementary Table 4) and an AceQ qPCR SYBR Green Master Mix Kit (Vazyme, Q111-02); mRNA of hamster $\beta$-actin was used as an internal control.

## Generation and validation of gene knock-outs

At least two sgRNAs were used per gene. The sgRNA sequences targeting *Sus scrofa LDLR* in LLC-PK1 or ST cell line, *Mesocricetus auratus Ldlr* or *Mxra8* in BHK-21 cell line are listed in Supplementary Table 4. Sequences encoding for sgRNAs targeting *Ldlr* were cloned into the plasmid lentiCRISPR v.2-puro (gift from Brett Stringer, Addgene plasmid #98290) and sequences encoding for sgRNAs targeting *Mxra8* were cloned into the plasmid lentiCRISPR v.2-neo (gift from Brett Stringer, Addgene plasmid # 98292). The HEK 293T cells grown in the 6-well cell culture plate were co-transfected with 2 µg obtained objective plasmids, 1.5 µg psPAX2 and 0.5 µg pMD2.0 G vectors using ExFect Transfection Reagent (Vazyme, T101). And 48 h post transfection the supernatants were collected and used to infect ST, LLC-PK1 or BHK-21 target cells. Clonal knockout cell lines were obtained by puromycin or neomycin selection and limited dilution method. Gene editing was confirmed with Sanger sequencing and western blot analysis.

## Virus binding and internalization assays

For virus-binding assays, monolayer of adherent cells grown in a 24-well cell culture plate to the density of 1 million cells per well was incubated with GETV-HN (MOI of 5) rSFV-mCherry (MOI of 5), rSINV-BEBV-mCherry (MOI of 5), rRRV-mCherry (MOI of 1), rSFV (MOI of 1) or rRRV (MOI of 1) at 4 °C for 30 min. After the removal of unbound virions, cells were washed three times with pre-cooled PBS and lysed in TRizol Up (TransGen Biotech, ET111-01-V2). For internalization assays, cells were grown, incubated with GETV-HN, and washed as described above. A pre-warmed (37 °C) DMEM supplemented with 2% FBS and 15 mM NH$_4$Cl was added and cells were incubated for 1 h at 37 °C to allow virus internalization. Then proteinase K at final concentration 500 ng ml$^{-1}$ was added and cells were incubated for 2 h at 4 °C to remove residual plasma membrane-bound virions. Finally, the cells were washed three times with pre-cooled PBS and lysed in TRizol Up reagent (TransGen Biotech, ET111-01-V2). For both assays, RNA extraction was performed using manufacturer's protocol and the cDNAs were obtained using Uni One-Step gDNA Removal and cDNA Synthesis SuperMix (TransGen Biotech, AU311). The qRT−PCR was performed using an AceQ qPCR SYBR Green Master Mix Kit (Vazyme, Q111-02) with mRNA of *gapdh* gene as an internal control.

## Confocal microscopy imaging of cells incubated with rGETV-mCherry virions

HEK 293T cells grown in a 15-mm glass-bottomed cell culture dish (Nest Biotechnology, China) were transfected with 1 µg of plasmid expressing hamster LDLR-Flag or LDLRAD3-Flag (control). At 24 h post-transfection cells were incubated with rGETV-mCherry (MOI of 200) at 4 °C or 37 °C for 20 min. The cells were washed three times with pre-cooled PBS, fixed with 4% PFA for 15 min at room temperature, permeabilized with 0.1% Triton X-100 for 5 min, and blocked with 5% skimmed milk powder in PBS at 37 °C for 1 h. Then, the cells were incubated for 2 h at room temperature with monoclonal anti-Flag M2 antibody (Sigma, F1804). After being washed three times with PBS, the cells were incubated for 10 min with FITC conjugated goat anti-mouse IgG Fc (Thermo Fisher, 31547, dilution 1:500). Fluorescence images were recorded using a Nikon A1 confocal microscope (Japan).

## Expression and purification of recombinant proteins

To obtain sequence encoding for GST-LBD fusion protein, cDNA fragment corresponding to amino acid residues 26-317 of the *Sus scrofa* LDLR extracellular domain was amplified by PCR and cloned into the pGEX-4T-1 vector (Cytiva #28-9545-49) between the *EcoRI* and *XhoI* sites. Obtained plasmid was verified by sequencing and used for transformation of BL21(DE3) *E. coli* cells. For expression of recombinant GST-LBD fusion protein cells were grown in Lysogeny Broth to an OD$_{600}$ of 0.8, induced with 0.5 mM IPTG for 12 h at 16 °C, harvested by centrifugation at 8000 x *g* for 10 min and resuspended in buffer containing 20 mM Tris-HCl [pH 8.0], 500 mM NaCl and 2 mM CaCl$_2$. Cells we lysed using sonication and lysate was subjected to centrifugation at 10,000 x *g* for 10 min. Obtained supernatant was incubated overnight with GST-tag purification resin (Beyotime, P2262) after which the resin was washed with equilibration buffer (200 mM NaCl, 20 mM Tris HCl [pH 8.0], 2 mM CaCl$_2$). Proteins bound to the resin (GST-LBD or GST used as a control) were then eluted with the same buffer supplemented with 15 mM glutathione. The eluted proteins were further concentrated using Amicon Ultra-15 30 K centrifugal filter devices (Millipore, UFC9030) and stored in Tris buffered saline (TBS [pH 7.4]) containing 2 mM CaCl$_2$.

For expression of recombinant protein in eukaryotic cells, DNA fragment corresponding to the coding sequence of *Mesocricetus auratus mxra8* gene or *Sus scrofa* LDLR extracellular domain was assembled from overlapping PCR fragments and cloned into the pcDNA3.4 vector (Thermo Fisher Scientific) in fusion with sequence encoding the human IgG Fc fragment using MultiF Seamless Assembly reagent (Abclone, RK21020). The obtained plasmid was verified by sequencing. 100 µg of plasmid DNA was used to transfect 100 ml suspension culture of Expi293F cells (density of 3 million cells ml$^{-1}$) using Poly-ethylenimine Linear (PEI) MW40000 transfection reagent (Yeasen Biotechnology, 40816ES03). Cells were cultivated for 4 days after which the cell culture supernatant was collected, filtered using 0.22 µm syringe filter (Millipore, SLGP033N), and protein A agarose (Beyotime, P2015) equilibrated with binding buffer (0.15 M NaCl, 20 mM Na$_2$HPO$_4$, pH 7.0) was added. Obtained mixture was incubated overnight at 4 °C, after which the supernatant was removed and recombinant protein was eluted from protein A agarose using 50 mM glycine buffer [pH 2.7]. Finally, the elution buffer was replaced with PBS through the use of Amicon Ultra-15 30 K centrifugal filter devices (Millipore, UFC9030); the purified protein was stored at -80 °C.

## Co-precipitation assay of virus or E2-E1 proteins with GST-LBD

The recombinant E2-E1 proteins of GETV, SFV, RRV and BEBV were obtained as follows. The sequence encoding E3-E2-6K-E1 (p62-E1) polyprotein, where 6 K region was replaced with sequence corresponding to four GGGGS repeats, was assembled from overlapping PCR fragments and inserted into pCAGGS vector in fusion with

sequence encoding for C-terminal HA tag. After sequence verification, 10 µg of resulting plasmid was used to transfect HEK 293T cells grown in a 60-mm glass-bottomed cell culture dish. At 24 h post transfection cells were lysed with NP-40 lysis buffer (Beyotime, P0013F), lysate was clarified using centrifugation at 12,000 x $g$ for 5 min and protein concentration in supernatant was measured using the bicinchoninic acid (BCA) protein assay kit (Vazyme, E112-01). Amounts of supernatant corresponding to 200 µg of total protein were used for immunoprecipitation.

Thirty µg of purified GST-LBD fusion protein (or GST, used as a control) was incubated with GST-tag binding resin for 2 h at 4 °C. After washing three times with DMEM, 50 µl GETV-HN stock containing $10^{6.2}$ $TCID_{50}$ of a virus or 20 µg GETV p62-E1-Strep or cell lysate, containing p62-E1 protein, was added. The resin was incubated for 2 h at 4 °C, washed, suspended in PBS and the appropriate amount of 4× SDS loading buffer containing β-mercaptoethanol was added. Samples were heated at 95 °C for 10 min and analyzed using SDS-PAGE and western blot.

To capture the GST-LBD fusion protein the assay was performed as follows. Twenty µg of 4D10 anti-E1 monoclonal antibody was incubated with protein A + G agarose (Beyotime, P2055) for 3 h at 4 °C. After washing the protein A + G agarose beads with DMEM, 50 µl of GETV-HN stock containing $10^{6.2}$ $TCID_{50}$ of a virus was added, and probes were incubated for 3 h at 4 °C. After washing three times with DMEM, agarose beads were incubated with 30 µg of purified GST-LBD fusion protein (or purified GST, used as control) for 2 h at 4 °C. Samples were prepared and analyzed as described above.

### Enzyme-linked immunosorbent assay (ELISA)
To evaluate in vitro binding of GETV virions to LBD of LDLR, 10 µg of 4D10 anti-E1 monoclonal antibody diluted in ELISA phosphate coating buffer (0.05 M carbonate buffer [pH 9.6]) was used to coat 96-well microtiter plate; coating was performed overnight at 4 °C. After three washes with PBS containing 0.001% Tween-20 (PBST), wells were blocked by incubation with 5% skimmed milk in PBS at 37 °C for 1 h. Then, GETV-HN virions ($10^5$ $TCID_{50}$ per well) diluted in PBS supplemented with BSA at final concentration 2% were added and plates were incubated for 1 h at room temperature. After three washes with PBST, different concentrations (0.1, 0.25, 0.5, 0.75, 1, 2.5, 5, 7.5, and 10 µg ml$^{-1}$) of GST (control) or GST-LBD proteins diluted in PBS containing 2% BSA and 2 mM $CaCl_2$ were added and the plates were incubated for 30 min at 37 °C. After extensive washes, wells were incubated with anti-GST polyclonal antibody (Proteintech, 10000-0-AP) diluted 1: 10,000 in PBS supplemented with 2 mM $CaCl_2$ for 2 h at room temperature. Next, after extensive washes, horseradish peroxide (HRP)-conjugated goat anti-rabbit IgG (KPL, 074-1806) diluted in PBS supplemented with 2 mM $CaCl_2$ was added followed by incubation for 1 h at room temperature. Finally, the plates were treated with 3,3'-5,5' tetramethylbenzidine substrate (Sigma, T4444) and then with 2 M $H_2SO_4$ to terminate the reaction. The absorbance at 450 nm was measured using automatic microplate reader (TECAN Infinite M200 Pro).

For antibody validation ELISA reactivity of mouse anti-LDLR or anti-MXRA8 polyclonal antibodies (pAbs) against GST-LBD or MXRA8-Fc recombinant proteins was analyzed. Purified proteins (50 µl, 5 µg ml$^{-1}$) were immobilized overnight at 4 °C on ELISA plates. Anti-LDLR, anti-MXRA8 and isotype control pAbs were added and plates were incubated for 1 h at room temperature. Signal was detected at 450 nm after incubation with HRP-conjugated goat anti-mouse IgG (H + L) and development with 3,3'-5,5' tetramethylbenzidine substrate.

### Production and purification of GETV p62-E1
The recombinant soluble GETV p62-E1-Strep was obtained as follows. The sequence encoding E3-E2-6K-E1 (p62-E1) polyprotein of GETV, where E1 and E2 transmembrane domains were removed and the 6 K

region was replaced with sequence corresponding to four GGGGS repeats, was assembled from overlapping PCR fragments and inserted into pcDNA3.4 vector in fusion with sequence encoding for C-terminal streptavidin binding tag. The 100 ml of suspension of Expi293F cells (density of 3 million cells ml$^{-1}$) was transfected with 100 µg of obtained plasmid using Polyethylenimine Linear (PEI) MW40000 reagent (Yeasen Biotechnology, 40816ES03). After 3 days of culturing the cell cultures supernatants were collected, filtered with 0.22 µm filter (Millipore, SLGP033N) and the recombinant protein was captured by incubation with Strep-Tactin XT 4Flow high-capacity resin (IBA, 2-5030-002) at 4 °C overnight. The resin was then washed with washing buffer and recombinant protein was eluted with BXT buffer (100 mM Tris/HCl, pH 8.0, 150 mM NaCl, 1 mM EDTA, 50 mM biotin). The buffer was replaced with PBS using 30 K Amicon filter columns (Millipore, UFC9030). The integrity and purity of GETV p62-E1-Strep was confirmed by SDS-PAGE. GETV p62-E1-Strep were not frozen, but stored at 4 °C.

### Production and purification of GETV virus-like particles (VLPs)
In order to produce VLPs of GETV, we cloned the DNA fragment encoding GETV structural polyprotein (capsid-E3–E2–6K–E1) into pcDNA3.4 vector. The 100 ml of suspension of HEK 293 F cells (density of 3 million cells ml$^{-1}$) was transfected with 100 µg of obtained plasmid using Polyethylenimine Linear (PEI) MW40000 reagent (Yeasen Biotechnology, 40816ES03). After 3 days of culturing the supernatants were collected, filtered with 0.22 µm filter (Millipore, SLGP033N) and incubated with PEG6000 at final concentration of 7% (w/v) and 2.3% NaCl (w/v) at 4 °C for overnight. The precipitated particles were collected by centrifugation at 4000 x g for 30 min, incubated on ice for 2 h and resuspended with PBS. The resuspended particles were loaded onto a 20–60% (w/v) continuous sucrose gradient and centrifuged at 160,000 x g for 1.5 h. The light scattering band corresponding to the GETV VLPs was collected and the buffer was replaced with that containing 20 mM HEPES [pH 8.0] and 150 mM NaCl using 30 K Amicon filter columns (Millipore, UFC9030). The integrity and purity of VLPs were confirmed by SDS-PAGE and negative-stain electron microscopy. VLPs were not frozen, but stored at 4 °C for biolayer interferometry binding assay.

### Biolayer interferometry (BLI) binding assay
The binding kinetics and affinity of GST-LBD fusion protein to GETV p62-E1-Strep or GETV VLPs were measured using an Octet RED96 (Pall Fortebio) instrument at 30 °C. Briefly, for binding GETV p62-E1-Strep, solution containing 10 µg ml$^{-1}$ GST or GST-LBD fusion proteins in the kinetic buffer (10 mM HEPES [pH 8.0], 150 mM NaCl, 1 mM $CaCl_2$, 0.02% Tween 20) was loaded onto GST biosensor (ForteBio, 18–5096) for 300 s. Then the biosensors were incubated with 150 nM of GETV p62-E1-Strep for 350 s to determine association kinetic. The biosensor was dipped into kinetics buffer for 100 s for measuring the dissociation kinetics. For binding GETV VLPs, 25 µg ml$^{-1}$ GST or GST-LBD fusion proteins in the kinetic buffer were loaded onto GST biosensor for 300 s. Then the biosensors were incubated with 150 nM of GETV VLPs for 600 s to determine association kinetic. The biosensor was dipped into kinetics buffer for 600 s for measuring the dissociation kinetics. Data analysis was performed with Data Analysis version 12.0 software (ForteBio).

### Preparation of anti-LDLR and anti-MXRA8 antibodies
For antibody preparation, 6-week-old BALB/c mice were immunized via subcutaneous route with 60 µg purified GST-LBD or MXRA8-Fc protein in complete Freund's adjuvant followed by two boosters with 30 µg GST-LBD or MXRA8-Fc protein in incomplete Freund's adjuvant with an interval of 14 days. Mice were euthanized and serum was isolated from bleedings. Sera were heat-inactivated at 56 °C for 30 min before performing the antibody inhibition assays, or stored at -80 °C

for use for western blot analysis. In some western blot and flow cytometry experiments, commercial antibodies were used because these experiments were performed before our own anti-LDLR antibodies become available.

## Entry blocking assays

To assess ability of GST-LBD fusion protein to inhibit GETV infection in vitro, rGETV-EGFP (MOI of 0.001 for BHK-21 cells, 0.1 for LLC-PK1, Vero or ST cells) was incubated with serial diluted GST-LBD or GST (control) proteins (0, 0.01, 0.05, 0.1, 0.5, 1, 5 or 10 µg ml$^{-1}$) for 30 min at 37 °C, and then added to the adherent cells in a 96-well cell culture plate ($5 \times 10^4$ cells per well). Cells were incubated for 1 h at 37 °C, washed three times with PBS and then supplemented with DMEM containing 2% FBS. At 18 h post infection, the percentage of EGFP-positive cells were measured by flow cytometry.

In another setup, polyclonal anti-LDLR mouse antiserum diluted in DMEM (dilutions 1:400, 1:200, and 1:100 for BHK-21 cells and 1:100 for ST, LLC-PK1 or Vero cells) was added to the adherent cells in a 96-well cell culture plate ($5 \times 10^4$ cells per well). After incubation for 30 min at 37 °C, cells were washed three times with PBS and infected with rGETV-EGFP (MOI of 0.001 for BHK-21 cells, 0.1 for LLC-PK1, Vero or ST cells) for 1 h at 37 °C, washed three times with PBS and then supplemented with DMEM containing 2% FBS. At 18 h post infection, infected cells were incubated and analyzed as described above.

## Mouse experiments

The two-day-old ICR mice ($n = 8$ per group) were inoculated subcutaneously with $10^5$ TCID$_{50}$ rGETV-HN before subcutaneous injection of 100 µg of GST-LBD fusion protein or GST protein (control). The survival rate and weight changes of mice were monitored daily for 15 days.

In another experiment two-day-old ICR mice ($n = 6$ per group) were infected and treated as described above. In addition, 6-week-old ICR mice ($n = 6$ per group) were inoculated subcutaneous with $10^4$ TCID$_{50}$ rGETV-HN before treatment with 700 µg GST-LBD fusion protein or GST protein (control) by intraperitoneal injection. Sera from 6-week-old infected and treated ICR mice were collected at 24 h and 48 h post infection. To measure the titer of virus in tissues, this subset of two-day-old mice was euthanized 6 days post infection while 6-week-old mice were euthanized 2 days post infection. Samples from spleen, lung or ankle were collected, 300 µl of PBS was added per 0.1 g of tissue and tissues were grind into homogenate. After centrifugation at 1000 x $g$ for 10 min aliquots of supernatants were collected and used to measure the viral titers.

All the mice were fed a 19% protein diet (Harlan Teklad, Irradiated), had 12 h light/dark cycle (0600-1800 h), and were housed in a facility maintained at a temperature range of 20-26 °C with a humidity range of 30–70%.

## Western blot

Cells were lysed by incubation with NP-40 lysis buffer (Beyotime, P0013F) for 30 min on ice. The protein concentration in the lysate was measured using the BCA protein assay kit (Vazyme, E112-01). An appropriate amount of 4× SDS loading buffer containing β-mercaptoethanol was added to an aliquot of the lysate and samples were heated at 95 °C for 10 min. Proteins were separated using SDS-PAGE in 10% gel, transferred onto blotting membranes and detected with protein- or tag-specific antibodies followed by incubation with HRP-conjugated secondary antibodies diluted in 5% solution of skimmed milk powder in PBS. Finally, proteins were visualized with an Amersham Imager 600. The following antibodies were used: anti-LDLR rabbit monoclonal antibody (ABclonal, A20808), anti-GST polyclonal antibody (Proteintech, 10000-0-AP), anti-mCherry (4C16) mouse

monoclonal antibody (Abmart M40012), anti-Flag M2 monoclonal antibody (Sigma, F1804), anti-HA monoclonal antibody (Sigma, H3663), anti-β-actin monoclonal antibody (Proteintech, 66009), anti-GAPDH monoclonal antibody (Proteintech, 60004), anti-MXRA8 or LDLR polyclonal mouse serum, GETV 3H2 anti-E1, 4D10 anti-E1, 8D5 anti-E2 monoclonal antibodies, and GETV anti-capsid protein polyclonal antibody (all in-house).

## Indirect immunofluorescence assay (IFA)

To quantify the relative infection of rGETV-EGFP, the infected cells were fixed with 4% (w/v) PFA for 30 min, and then permeabilized with 0.1% Triton X-100 in PBS for 20 min. After washing three time with 0.01% Triton X-100 in PBS, cells were incubated with DAPI (Solarbio, 1:1000) for 10 min prior to observing using a differential fluorescence microscope (Nikon); two-color fluorescence images were recorded from at least three separate experiments.

## Evaluation of cell viability

Cells were plated into a 96-well cell culture plate at a density of $2 \times 10^4$ cells per well. After 24 h incubation at 37 °C, 10 µl of solution of Cell Counting Kit 8 (CCK8; Beyotime, C0038) was added to each well. Plates were incubated for 1 h at 37 °C, then the absorbance at 450 nm was determined with automatic microplate reader (TECAN Infinite M200 Pro).

## Plaque assays

BHK-21 cells grown on 24 well cell culture plates were infected with rGETV-HN, rGETV-mCherry, rGETV-EGFP, rSFV-mCherry, rRRV-mCherry or rSINV-BEBV-mCherry for 1 h at 37 °C, washed three times with PBS to remove unbound virions, and then covered with a mixture (1:1) of 2% solution of methylcellulose and 2× DMEM with 4% FBS for 36 h. After washing three times with PBS cells were stained with crystal violet for 30 min at room temperature, thoroughly washed with running water, and dried for imaging.

## Statistical analysis

Data are presented as the mean $\pm$ standard deviation (SD) from three or more independent experiments. Data were analyzed using Graph-Pad Prism 9. The significance of variability between groups was determined by Student's t-test. Comparison of survival curves was performed by log-rank test. $P$-values < 0.05 were considered as statistically significant.

## Reporting summary

Further information on research design is available in the Nature Portfolio Reporting Summary linked to this article.

# Data availability

All data generated in this study, which include original data and images, are provided in the Supplementary Information. Source data are provided with this paper.

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

## Acknowledgements

This work was financially supported by the National Key Research and Development Program of China (grant no. 2022YFC2604203 to S.S.); The 2021 Agricultural Research Outstanding Talents Training Program of the Ministry of Agriculture and Rural Affairs (S.S.); The Young Top-Notch Talents of National 10 Thousand Talent Program and the Bioinformatics Center of Nanjing Agricultural University (S.S.); The Guangzhou Science and Technology Plan Project (202103000029 to Y.M.); The National Natural Science Foundation of China (91853131, 81872786 to Y.M.). We thank Zhiyu Shi from the instrument platform of Institute of Immunology, College of Veterinary Medicine, Nanjing Agricultural University, for assistance in using the differential fluorescence microscope.

## Author contributions

All authors contributed equally to this work. S.S. conceived the idea and oversaw the project. S.S., Y. M., X.Zhai. designed the experiments and analyzed the data. X.Zhai, X.L. performed the experiments. S.S., Y.M., X.Zhai, X.L., M.V. prepared the figures and wrote the manuscript. Y.W. mainly expressed recombinant protein and prepared antibody. N.W. participated and repeated part of the experiments. S.S., Y.M., X.Zhai, X.L., M.V., A.M. reviewed the manuscript. Z.J. participated part of the mouse experiments. X.Zhang. helped FACs analysis and infection assay. Y.Q., K.Q., H.J. did the western blot analysis. W.H. Y.C. critically reviewed the study. All authors discussed the results and commented on the manuscript.

## Competing interests

The authors declare no competing interests.
