## [Peer Review File · Nature Communications]

LDLR is used as a cell entry receptor by multiple alphavirusesREVIEWER COMMENTS

Reviewer #1 (Remarks to the Author):

Zhai et al., in their manuscript entitled "LDLR is used as a cell entry receptor by multiple alphaviruses," report on the identification of low-density lipoprotein receptor (LDLR) as a cellular entry receptor for Getah virus (GETV), Semliki forest virus (SFV), Bebaru virus (BEBV), and Ross River virus (RRV). They first demonstrated that LDLR promotes GETV entry using an eGFP-tagged library of membrane proteins. In these assays, they used a genetically modified GETV that contains an internal mCherry tag, and LDLR was a top hit. They then used overexpression studies in HEK293T cells to show that GETV lentivirus-based pseudotype entry is enhanced by LDLR overexpression on HEK293T cells. Interestingly, the entry of SFV, BEBV, and RRV pseudotypes was also enhanced.

The authors then carried out extensive work to demonstrate that LDLR binds directly to the E2-E1 spike proteins of GETV. This work included ELISAs and biolayer interferometry (BLI) experiments with virus-like particles. They also map the interaction as likely involving the cysteine-rich (CR) repeats 4 and 5 in the ligand binding domain of LDLR. Additional experiments included showing that exogenous addition of soluble receptor or anti-LDLR antibody blocks viral entry, and that the receptor likely participates in GETV virus binding to cells and internalization. Lastly, they show that soluble receptor can protect mice from lethal challenge with GETV. Overall, the work demonstrating that LDLR is a likely cellular receptor for GETV is well executed and properly controlled, and the findings with this virus are compelling. However, this reviewer has major concerns about the work implicating LDLR as a cellular receptor for SFV, BEBV, and RRV, as this work was only done with one assay and relied on lentivirus pseudotypes that may have improperly assembled spike proteins. Thus, while the work implicating LDLR as a receptor for GETV is complete, the work implicating LDLR as a receptor for SFV, BEBV, and RRV is too preliminary and falls short of the standard used to define the role of specific membrane proteins as cellular receptors for viruses.

MAJOR:

1. The data in Figure 1d-k were generated with lentivirus pseudotypes. These lentivirus pseudotypes are unlikely to contain properly assembled E2/E1 protomers (with icosahedral assembly). So, while this experiment is well controlled (e.g., expected phenotypes are seen with Mxra8 overexpression with chikungunya virus pseudotypes), it remains critical to also use a different system that faithfully recapitulates alphavirus spike protein organization to make a claim on receptor use for SFV, BEBV, and RRV (e.g., chimeric Sindbis viruses, which are the standard in the field, or authentic alphaviruses).

For example, a separate study found only low levels of entry of alphavirus replicon-based SFV reporter virus particles with cells overexpressing LDLR; but a much more significant effect was seen with VLDLR overexpression (PMID: 34929721). Structural analysis of VLDLR bound to SFV virus-like particles also suggests that there may be important requirements for receptor assembly near icosahedral axes of the E2-E1 spike proteins (PMID: 37098345). It is thus possible that sites that might be exposed for receptor binding by LDLR on SFV-lentivirus pseudotypes might not be exposed on properly assembled E2-E1 spikes in the context of native or native-like virions.

2. Critical assays for SFV, BEBV, and RRV are missing to conclude that LDLR is a receptor for these viruses. These assays include:

- (a) Showing that LDLR binds to the E2-E1 proteins of these viruses through immunoprecipitation experiments, ELISAs, or BLI experiments.
- (b) Assays showing that LDLR expression results in binding to cells and virus internalization.
- (c) As noted above, assays with authentic viruses are missing.

3. VLDLR, a known alphavirus receptor that is very closely related to LDLR and is also a receptor

for multiple alphaviruses including SFV, was not tested anywhere in the manuscript. VLDLR (and ApoER2) are two canonical members of the LDLR family that have similar structural organization and sequence homology in their ligand-binding domains to LDLR (while LDLRAD3, which the authors tested, is quite different). While the manuscript centers on LDLR, interest would be significantly increased if the authors demonstrated that LDLR but not VLDLR or ApoER2 are cellular receptors for GETV. This finding would have important implications to the general understanding of the evolution of alphavirus receptor specificities.

In a related question, were VLDLR and ApoER2 on the list of the 150 membrane proteins tested in their screen? The list of membrane proteins should be provided in the supplemental materials for completeness.

4. Biolayer interferometry assays in Figure 3i. The authors dipped sensor tips that were coated with receptors into solution containing VLPs, and report measuring an affinity of 263 nM. There would be substantial avidity in the system, which would falsely decrease the off rate. Any KD determination would therefore be uninterpretable. For affinity measurements, the virus would have to be immobilized on the sensor tip and the receptor kept in solution to minimize avidity.

5. Antibody blocking assays in Figure 4 e-h rely on mouse anti-LDLR serum, which may have some cross-reactivity with other LDLR family members (e.g., with VLDLR) and confound the results of this assay. Could the authors comment on how this antibody was validated, and is it possible that it may cross-react with VLDLR, LRP1, etc.?

6. For the confocal microscopy studies (Figure 2k), how many images were used to quantify binding? Could additional images be provided in the supplement?

7. Abstract: "Although several receptors have been identified for alphavirus entry, they could not explain broad host range and multi-tissue tropism of alphaviruses, indicating the existence of additional receptors." This reviewer does not agree with this statement. Another, very closely related LDLR-related family receptor, VLDLR, as already been described as a conserved receptor for SFV, and orthologs from vertebrates and invertebrates are functional receptors. Would recommend rephrasing to say that some of the previously identified receptors cannot explain the broad host range and tissue tropism of certain alphaviruses.

Minor:

Lines 70-71: "However, ApoER2 and VLDLR are almost exclusively expressed in the central nervous system" While ApoER2 expression is mostly limited to the brain, VLDLR is more widely expressed, with expression in the heart, muscle, lung, kidneys, and adipose tissue. See reference PMID: 21554715.

Figure 1: HEK293T cells are used for overexpression studies, but how much native expression of LDLR is there on these cells?

Extended data Figure 1b. Additional molecular weight marker labeling should be provided.

Line 158: Could the authors define ST and LLC-PK1 cells regarding their relevance?

Figures 4a-b. Statistics should be shown.

Extended Data 7: It would be useful for the authors to provide a schematic diagram describing how the LDLR-deltaCR5+LDLRAD3-D1 construct was designed.

Reviewer #2 (Remarks to the Author):

This manuscript by Zhai et al. describes a test of 150 membrane proteins for roles in promoting infection by Getah Virus. GETV is an alphavirus that is an important pathogen of pigs, and knowledge of its receptor is important in understanding infection and pathogenesis. The paper reports that LDLR is a receptor for GETV, and also for SFV, RRV, and BEBV. There have been in the last few years a number of very strong papers on alphavirus receptors, including both functional and structural studies: MXRA8 for CHIKV, LDLRAD3 for VEEV, VLDLR and ApoER2 for SFV and EEEV. Given these extensive studies, the bar to prove the role of a new alphavirus receptor is high. The current paper represents a considerable amount of work in testing viruses, testing various receptor constructs, mapping binding sites, and performing infection/blocking experiments in mice. However, there are serious concerns about the constructs and some other reagents used in the paper, and these issues need to be addressed thoroughly before the reader can be convinced of the importance of LDLR as a receptor for GETV and also for SFV, RRV, and BEBV.

Major points:

1. The paper suffers from a serious deficiency of careful documentation of the various constructs used. The library of 150 membrane proteins are tagged with eGFP—there is no documentation of whether the tag is on the N- or C-terminus, and whether it affects delivery of the protein to the membrane. More importantly, for the N-tagged LDLR constructs, there is no information on whether the FLAG tag is placed after the signal sequence, and if not, how the protein would be inserted into the ER. The studies lean heavily on the use of a soluble LDL LBD-GST protein, which is used to block infection in cell culture and in mouse studies, and to produce an antibody that is used in cell blocking experiments. This LBD-GST protein is produced in bacteria. There is no documentation that this heavily disulfide-bonded protein is folded correctly, or references to published work that might support this. Might this explain why the western blot and flow cytometry studies use commercial Abs to LDLR or FLAG, rather than the polyclonal antibody they generated?
2. It is critical to confirm that the various LDLR mutants are expressed on the cell surface. This is addressed by flow cytometry and receptor staining. The receptors are described to be N-terminally tagged i.e. the FLAG tag should be localised extracellularly. However, the flow cytometry methods state that cells were permeabilised to detect the FLAG tag. From this they then conclude the receptor cell surface levels. This is not accurate as permeabilisation will allow the entire receptor population to be labelled (i.e both cytoplasmic and PM localised). Without cell surface confirmation of all receptor constructs, the data are not conclusive. (The methods also read like cells are fixed prior to trypsinisation, which would prevent the cells from coming off the plate for the flow analysis).
3. The endocytosis assays in Fig. 2 are not convincing. There is no stripping step to remove bound virus after uptake (2J). The fluorescence figure in 2K shows binding of virus, but what is needed in addition is to demonstrate that the receptor promotes uptake (e.g., this was shown very convincingly in the Clark et al paper). Such experiments are critical to prove that LDLR is actually acting as a receptor. The discussion of the role of receptor in virus endocytosis should cite the highly relevant paper by Feng et al, *J. Virol* 2023, and also the receptor studies (MXRA8, LDLRAD3) that examined the role of the TM and cytoplasmic tail.
4. Another key point is to demonstrate that GETV p62-E1 binds directly to the LDLR, ruling out other proteins acting as intermediaries. However, the p62-E1 construct that is produced in 293 cells was not purified for these studies, but a cell lysate prep was used. While the result may be correct, it would be much more convincing to demonstrate this with purified proteins. The GETV VLP studies look more convincing, but the authors should consider whether associated LDL molecules might explain the interaction (as was considered in the Clark et al paper).
5. The sequence comparisons in extended data Fig. 8 are interesting and used to identify candidate sites of interaction (see also Fig. 6). However, these comparisons would be much stronger if the permissive CR 4 and CR5 domains were compared to the sequences of the non-permissive CR 1, 2 etc. In ED Fig 8, the comparison should include non-permissive host LDLR, and the mosquito homologue.
6. Fig 2n presents the rather surprising finding that the deletion of MXRA8, which decreases GETV infection, can be rescued by expression of LDLR. What about the converse experiment---can the

LDLR deletion or the LDLR/MXRA8 double deletion be rescued by MXRA8? The result currently is presented without any discussion of what it might mean.

Minor points:

1. Please briefly describe the construction of the rGETV-eGFP reporter virus, which does not seem to be in ref. 33. Is this a cytoplasmic reporter? If the construct is similar to that of rGETV-mCherry, is there a problem with the cysteines in eGFP? Does the virus grow efficiently?
2. In the studies of the KO cells in extended data Fig. 4, why are there not results for both alleles of the various proteins? It would be quite surprising if both alleles had the same CRISPR-edited sequence for all of the KO. Please also explain if these are cell clones or populations.
3. Throughout the paper, labeling of the constructs should accurately indicate the position of the tag. For example, the LDLR constructs state that they have N-terminal FLAG tags, but are designated LDLR-FLAG. This is confusing and not the usual convention.
4. Line 66. Please adjust text to reflect that the mode of interaction (ie structural analysis) has also been determined for MXRA8 and LDLRAD3.
5. Line 71 is inaccurate. Both NCBI and Human protein atlas databases report very wide tissue expression of at least human VLDLR.
6. Please adjust text in introduction to reflect that the Clark et al paper tested VLDLR from many species including mosquito.
7. It's very hard to see plaques in ED Fig 1e. Please fix figure.
8. Why does LDLR-FLAG show two bands in ED Fig 1g?

Reviewer #3 (Remarks to the Author):

In this study, the authors identified LDLR as a crucial receptor involved in entry of several viruses belonging to the Alphavirus genus into target cells, whereby Getah virus was the main focus of the study due to biological safety reasons. The overall experimental strategy was well-designed. However, there is major concern if LDLR is indeed the crucial receptor for these viruses without validation experiments with in-vivo knockout model of LDLR. In-vivo model of LDLR knockout are well established and has been utilized for diseases such as hypercholesterolemia.

1) The authors have constructed an expression plasmid library for 150 mouse membrane proteins, in order to identify the specific protein(s) which positively affect alphavirus infections. What are the selection criteria of these proteins, have these proteins been shown in the past to be associated with the infection capacity of other viruses?

2) It will be useful for the authors to provide a list of the tested proteins including those which were negative hits in this study, as it can serve as an important source of information for future works by the readers.

3) Figure 2j: Can the authors provide a clear description on the y-axis of the graph? Is the y-axis presented as relative ratio of internalization:binding data? The error bars are too huge in my opinion to deduce that there is an effect on the virus internalization, while there are some data points which showed similar levels of vRNA as the WT.

4) In Figure 2t, the authors showed that JEV is among the virus which are not affected by LDLR deficiency in the host cells. However, this is in contrast to a previous finding (<https://www.ncbi.nlm.nih.gov/pmc/articles/PMC8238074/>) which mechanistically demonstrated that LDLR is an important host factor required for JEV entry. The authors will need to address this contrasting data.

5) Although the mouse sera used in this study was collected from mice subcutaneously injected

with GST-LBD, how did the author confirm the presence of anti-LDLR antibodies in the sera generated from their mouse model? Was there any neutralization test being performed using the generated sera? The authors need to convincingly show that the inhibition observed on the rGETV-EGFP infection within the different cell lines tested are indeed due to the presence of anti-LDLR antibodies.

6) It is not convincing with the rather low levels of infection efficiency by the different viruses generated by the authors in Figure 1, it is perplexing with the extremely low levels of EGFP+ cells illustrated in Figures 3d, 6b and 6g, within the WT or empty vector control groups, which are rather contrasting and not even close to the initial data in Figure 1d showed an approximately 10% infection by GETV. Another concern from this point is that the rather low infection levels of the WT group might result in a biased over-interpretation of the efficiency of LDLR expression in the rest of the experiments in this study.

7) Validation experiments of virus infectivity and pathogenesis with in-vivo knockout model of LDLR is essential.

7) Figure 7b and c: It is inconclusive for the authors to deduce that the CR4 region are also essential for promoting GETV infection as there were still statistically significant upregulation of the EGFP+ cells and viral titers in the absence of CR4.

8) Line 223: Was the promoting effect in the absence of CR4 statistically compared to CR5? It is inaccurate for the authors to state that "deletion of CR4 statistically reduced the promoting effect of LDLR expression", as the statistics performed were on the upregulation of EGFP+ cells and viral titers relative to the WT group.

Minor comments:

1) Figure 1b: The authors should add another sub-section on the lower end of y-axis, as the current graphical illustration gives a rather confusing perspective that the vector and LDLRAD3 groups do not generate any viral titer whereas the actual scenario is that the vector has a 100% relative viral RNA level.

2) Figure legend for Figure 6: PDB IDs have to be in uppercase.

3) There are a few typing errors across the manuscript

Reviewer #1 (Remarks to the Author)

Major:

1. The data in Figure 1d-k were generated with lentivirus pseudotypes. These lentivirus pseudotypes are unlikely to contain properly assembled E2/E1 protomers (with icosahedral assembly). So, while this experiment is well controlled (e.g., expected phenotypes are seen with Mxra8 overexpression with chikungunya virus pseudotypes), it remains critical to also use a different system that faithfully recapitulates alphavirus spike protein organization to make a claim on receptor use for SFV, BEBV, and RRV (e.g., chimeric Sindbis viruses, which are the standard in the field, or authentic alphaviruses).

For example, a separate study found only low levels of entry of alphavirus replicon-based SFV reporter virus particles with cells overexpressing LDLR; but a much more significant effect was seen with VLDLR overexpression (PMID: 34929721). Structural analysis of VLDLR bound to SFV virus-like particles also suggests that there may be important requirements for receptor assembly near icosahedral axes of the E2-E1 spike proteins (PMID: 37098345). It is thus possible that sites that might be exposed for receptor binding by LDLR on SFV-lentivirus pseudotypes might not be exposed on properly assembled E2-E1 spikes in the context of native or native-like virions.

Response:

Authors thank the reviewer for the comment and suggestion. We understand the concern for the organization of E2-E1 on HIV pseudo-viruses, which might affect the interaction with LDLR. However, we would like to point out that our preliminary model of this interaction does not rely on the organization of E2-E1 on virus particles (Fig. 6e of the main text). Nevertheless, we have now constructed recombinant reporter viruses of rSFV-mCherry, rRRV-mCherry, and rSINV-BEBV-mCherry (chimeric Sindbis viruses). Binding and internalization experiments indicated that all three recombinant viruses had increased infection in the presence of LDLR overexpression, in a similar way to our previous results with HIV pseudo-viruses. Please see the Extended Data Fig. 6. In addition, the extent of each recombinant reporter viruses benefiting from LDLR followed the trend displayed in HIV pseudo-virus experiments (Fig. 1c-k of the main text). For example, RRV benefits less from overexpression of LDLR than SFV, GETV and BEBV in both assays. These new data reinforced our initial conclusion with HIV pseudo-viruses. The construction and validation of these recombinant viruses were put into Extended Data Fig. 1.

Extended Data Fig. 6

2. Critical assays for SFV, BEBV, and RRV are missing to conclude that LDLR is a receptor for these viruses. These assays include:

(a) Showing that LDLR binds to the E2-E1 proteins of these viruses through immunoprecipitation experiments, ELISAs, or BLI experiments.

(b) Assays showing that LDLR expression results in binding to cells and virus internalization.

(c) As noted above, assays with authentic viruses are missing.

Response:

Thank you for the suggestions. In the response above, we have provided new data of binding and internalization experiments using recombinant viruses of rSFV-mCherry, rRRV-mCherry, and rSINV-BEBV-mCherry. In addition, we have now included the data from immunoprecipitation experiments with HA-tagged envelope proteins of SFV, BEBV, and RRV. As shown below, GST-LBD protein can interact with p62-E1-HA of SFV, BEBV, and RRV. The new IP data is added to Extended Data Fig. 9a-c.

Extended Data Fig. 9

3. VLDLR, a known alphavirus receptor that is very closely related to LDLR and is also a receptor for multiple alphaviruses including SFV, was not tested anywhere in the manuscript. VLDLR (and ApoER2) are two canonical members of the LDLR family that have similar structural organization and sequence homology in their ligand-binding domains to LDLR (while LDLRAD3, which the authors tested, is quite different). While the manuscript centers on LDLR, interest would be significantly increased if the authors demonstrated that LDLR but not VLDLR or ApoER2 are cellular receptors for GETV. This finding would have important implications to the general understanding of the evolution of alphavirus receptor specificities.

In a related question, were VLDLR and ApoER2 on the list of the 150 membrane proteins tested in their screen? The list of membrane proteins should be provided in the supplemental materials for completeness.

Response:

Authors thank the reviewer for this good suggestion. We have tested the effects of VLDLR and ApoER2 (LRP8) on GETV. As shown below, indeed, VLDLR and ApoER2 can also significantly promote GETV infection in HEK 293T. In addition, to analyze whether LDLR can act independently as receptor we overexpressed LDLR in K562 cells lacking VLDLR and LRP8 (DOI: 10.1038/s41586-021-04326-0), and found that LDLR could significantly promoted viral infection. This suggests that these three closely related receptors, which belong to the LDLR receptor family

and contain CR repeats, can all act as receptors of one and the same alphavirus. Interestingly, according to previous research, VLDLR CR3 binds SFV E1-DIII (DOI: [10.1016/j.cell.2023.03.032](https://doi.org/10.1016/j.cell.2023.03.032)), while our data shows that for LDLR CR4, CR5 can also mediate GETV infection. There seems to be a general rule for alpha viruses and LDLR receptor family. However, it requires much more work to substantiate this theory and extend it to all representative alpha viruses. In our opinion, comprehensive analysis of the relationships between LDLR receptor family proteins and alphaviruses falls beyond the scope of this study. For this reason, we decided not to include the (compared to the extensive analysis performed with LDLR) rather limited data of VLDLR and LRP8 to the manuscript. Nevertheless, it represents an attractive topic for future studies.

In addition, it should be noted that VLDLR and ApoER2 were not in the list of the 150 membrane proteins analyzed in this study; so the fact that they were not initially discovered as receptors for GETV is due to the limited size of our library. The list of membrane protein present in the library has now been provided in the Extended Data Table 1.

4. Biolayer interferometry assays in Figure 3i. The authors dipped sensor tips that were coated with receptors into solution containing VLPs, and report measuring an affinity of 263 nM. There would be substantial avidity in the system, which would falsely decrease the off rate. Any KD determination would therefore be uninterpretable. For affinity measurements, the virus would have to be immobilized on the sensor tip and the receptor kept in solution to minimize avidity.

Response:

Authors agree that coating the sensor with VLPs will have less avidity issue. However, we do want to point out that there are also many publications that did the experiments in the reverse way (DOI: [10.1038/s41586-021-04326-0](https://doi.org/10.1038/s41586-021-04326-0); DOI: [10.1002/biot.201800303](https://doi.org/10.1002/biot.201800303); DOI: [10.1126/sciadv.abl6015](https://doi.org/10.1126/sciadv.abl6015); DOI: [10.1128/JVI.01871-16](https://doi.org/10.1128/JVI.01871-16) DOI: [10.1007/s00216-015-8735-x](https://doi.org/10.1007/s00216-015-8735-x)). That being said, in order to completely rule out the avidity issue, we replaced VLP with purified p62-E1 protein, and re-did BLI experiment. As shown in the figure below, we recorded a KD value of 273 nM in this way. We added this new data to Extended Data Fig. 8d.

Extended Data Fig. 8d

5. Antibody blocking assays in Figure 4 e-h rely on mouse anti-LDLR serum, which may have some cross-reactivity with other LDLR family members (e.g., with VLDLR) and confound the results of this assay. Could the authors comment on how this antibody was validated, and is it possible that it may cross-react with VLDLR, LRP1, etc.?

Response:

Thanks for your suggestion, we have now included validation data of anti-LDLR serum. Briefly, it is found that this polyclonal anti-LDLR serum recognize GST-LBD and LDLR-Flag expressed in HEK 293T cells but do not recognize VLDLR-Flag. This new data is now included in Extended Data Fig. 11 of the revised manuscript.

6. For the confocal microscopy studies (Figure 2k), how many images were used to quantify binding? Could additional images be provided in the supplement?

Response:

We apologize for lack of details in the image analysis. As suggested by other reviewers, we re-did the experiment with a higher dose of the virus (MOI=200). This new data is now displayed in Fig. 2b. For image analysis, we quantified the number of virions in each experimental group using three individual images taken with full field-of-view (as shown below). ImageJ software was set to randomly pick up 75 cells for quantification in each experimental group. And a representative zoom-in image was put into the figure for illustration purpose. We will provide all the full field-of-view images used for quantification in a raw data file to the manuscript.

7. Abstract: “Although several receptors have been identified for alphavirus entry, they could not explain broad host range and multi-tissue tropism of alphaviruses, indicating the existence of additional receptors.” This reviewer does not agree with this statement. Another, very closely related LDLR-related family receptor, VLDLR, has already been described as a conserved receptor for SFV, and orthologs from vertebrates and invertebrates are functional receptors. Would recommend rephrasing to say that some of the previously identified receptors cannot explain the broad host range and tissue tropism of certain alphaviruses.

Response:

Thanks for pointing this out. We have changed the sentence to " So far several receptors have been identified for alphavirus entry, however, they cannot explain the broad host range and tissue tropism of certain alphaviruses, such as Getah virus (GETV), indicating the existence of additional receptors." Please see lines 21-24.

Minor:

Lines 70-71: “However, ApoER2 and VLDLR are almost exclusively expressed in the central nervous system” While ApoER2 expression is mostly limited to the brain, VLDLR is more widely expressed, with expression in the heart, muscle, lung, kidneys, and adipose tissue. See reference PMID: 21554715.

Response:

Thanks for pointing this out. We have changed the sentence (lines 77-81) to "Similarly, for SFV, ApoER2 is almost exclusively expressed in the central nervous system. Although VLDLR is expressed in multiple tissues, it has been observed that the absence of VLDLR or presence of antibodies against VLDLR do not completely block the infection of SFV in Vero, U20S, A549, Huh7 and other cells".

Figure 1: HEK293T cells are used for overexpression studies, but how much native expression of LDLR is there on these cells?

Response:

Please refer to the Western Blots shown below for the level of native expression of LDLR of HEK 293T in comparison to overexpression of Flag-tagged LDLR.

Extended data Figure 1b. Additional molecular weight marker labeling should be provided.

Response:

Thank you very much for your comments. We have amended the figure with additional molecular weight markers and due to other revisions, it is now in Extended Data Fig. 1c.

Line 158: Could the authors define ST and LLC-PK1 cells regarding their relevance?

Response:

A previous study demonstrated the recovery of GETV from various organs of experimentally infected piglets, including testis and kidney. GETV was detected in all tissues collected from one of the piglets. (DOI: 10.1292/jvms1939.49.1003). In addition, infection of porcine kidney cells produced rapid cytopathic effects (CPEs), including shrinking, rounding and detaching, and peak titers of $10^{9.3}$ TCID₅₀/ml was measured at 24 h post-infection. (<https://doi.org/10.1111/tbed.13567>). Therefore, we selected two of the few cell lines available from pigs, common ST (porcine testicular cells) and LLC-PK1 (porcine kidney cells) for our experiments. To clarify, we added the introduction to GETV (lines 87-94) and description of the relevance of the two pig cell lines (lines 204-206) in the revised manuscript.

Figures 4a-b. Statistics should be shown.

Response:

Thanks for pointing this out. The statistics is now shown in Fig. 4a-b.

Extended Data 7: It would be useful for the authors to provide a schematic diagram describing how the LDLR-deltaCR5+LDLRAD3-D1 construct was designed.

Response:

Thank you for your suggestion. We have now made a diagram for LDLRAD3- Δ D1+LDLR-CR5 and LDLR- Δ CR5+LDLRAD3-D1, and included it in Extended Data Fig. 12a.

Reviewer #2 (Remarks to the Author):

Major points:

1. The paper suffers from a serious deficiency of careful documentation of the various constructs used. The library of 150 membrane proteins are tagged with eGFP—there is no documentation of whether the tag is on the N- or C-terminus, and whether it affects delivery of the protein to the membrane.

Response:

We sincerely apologize for the lack of detailed information of the constructs used in this study. First of all, to make it clear, plasmids in the library of membrane proteins were constructed to have EGFP as a co-expressing gene, instead of a fusion-tag. Please see the plasmid map below. The co-expression of EGFP as a separate protein in theory does not affect the localization of the membrane proteins in the library. We amended the Methods with the details of plasmid construction (lines 416-419). Nevertheless, we further confirmed the localization of the membrane protein hits in our study by laser scanning confocal microscopy (TIMD4 and LDLR).

More importantly, for the N-tagged LDLR constructs, there is no information on whether the FLAG tag is placed after the signal sequence, and if not, how the protein would be inserted into the ER.

Response:

Actually, the constructs for LDLR proteins were designed to have Flag tag in the C terminus, which was described in lines 449-453 of the Methods but was put in a wrong location in Fig. 3a of our last draft. We sincerely apologize for the confusion caused by our carelessness. We have now corrected Fig. 3a to reflect the correct location of the Flag tag in our revised manuscript.

The studies lean heavily on the use of a soluble LDL LBD-GST protein, which is used to block infection in cell culture and in mouse studies, and to produce an antibody that is used in cell blocking experiments. This LBD-GST protein is produced in bacteria. There is no documentation that this heavily disulfide-bonded protein is folded correctly, or references to published work that might support this.

Response:

First of all, it has been demonstrated that soluble *E.coli* expressed GST-CR domains can functionally block VSV infection (doi: 10.1038/s41467-018-03432-4). In addition, for further proof, we have now provided comparative analysis of *E.coli* expressed GST-LBD with a mammalian cell (Expi293F) expressed LBD-Fc in blocking virus infection. As shown below, GST-LBD and LBD-Fc block GETV infection to a similar extent, which confirms our *E.coli* expressed GST-LBD is functional. We have added this to Extended Data Fig. 10.

Extended Data Fig. 10

Might this explain why the western blot and flow cytometry studies use commercial Abs to LDLR or FLAG, rather than the polyclonal antibody they generated?

Response:

Commercial antibodies were used to in western blot and flow cytometry simply because these experiments were performed before our own anti-LDLR were available.

2. It is critical to confirm that the various LDLR mutants are expressed on the cell surface. This is addressed by flow cytometry and receptor staining. The receptors are described to be N-terminally tagged i.e. the FLAG tag should be localized extracellularly. However, the flow cytometry methods state that cells were permeabilised to detect the FLAG tag. From this they then conclude the receptor cell surface levels. This is not accurate as permeabilisation will allow the entire receptor population to be labelled (i.e both cytoplasmic and PM localised). Without cell surface confirmation of all receptor constructs, the data are not conclusive. (The methods also read like cells are fixed prior to trypsinisation, which would prevent the cells from coming off the plate for the flow analysis).

Response:

We thank author for the comment and apologize again for the confusion caused by grammatical errors and inaccurate description. As mentioned above, the constructs for LDLR proteins were in fact designed to have Flag tag at the C terminus. That was the reason why we did permeation before staining the cells. We have now used confocal microscopy to confirm the membrane localization of overexpressed LDLR mutants and added this data to Extended Data Fig. 4b.

Extended Data Fig. 4b

3. The endocytosis assays in Fig. 2 are not convincing. There is no stripping step to remove bound virus after uptake (2J). The fluorescence figure in 2K shows binding of virus, but what is needed in addition is to demonstrate that the receptor promotes uptake (e.g., this was shown very convincingly in the Clark et al paper). Such experiments are critical to prove that LDLR is actually acting as a receptor. The discussion of the role of receptor in virus endocytosis should cite the highly relevant paper by Feng et al, *J. Virol* 2023, and also the receptor studies (MXRA8, LDLRAD3) that examined the role of the TM and cytoplasmic tail.

Response:

Thanks for pointing this out. Actually, we did use proteinase K to remove non-internalized virions for the qPCR-based endocytosis assays (lines 523-526 in the Methods). We chose this approach because it was used in two relevant papers. (DOI: 10.1038/s41586-018-0121-3 DOI: 10.1038/s41586-020-2915-3). Nevertheless, we have now followed the reviewer's suggestion and provided additional evidence of LDLR-promoted cell surface binding and internalization of rGETV using confocal microscopy as in the Clark et al paper. The new data has been included in Fig. 2b of the revised manuscript. As is demonstrated in this figure, cells overexpressing LDLR protein were incubated with rGETV (MOI=200) for 20 min at 4°C or 37°C, and imaged for co-localization of virus particle and LDLR. The results confirmed that LDLR-promoted cell surface binding of virus particles at 4°C and their internalization at 37°C, reinforcing our conclusion of LDLR-promoted virus uptake. In addition, we now cite the highly relevant paper by Feng et al, *J. Virol* 2023, please see reference 18.

Fig. 2b

4. Another key point is to demonstrate that GETV p62-E1 binds directly to the LDLR, ruling out other proteins acting as intermediaries. However, the p62-E1 construct that is produced in 293 cells was not purified for these studies, but a cell lysate prep was used. While the result may be correct, it would be much more convincing to demonstrate this with purified proteins. The GETV VLP studies look more convincing, but the authors should consider whether associated LDL molecules might explain the interaction (as was considered in the Clark et al paper).

Response:

Authors thanks the reviewer for the suggestion. We have now included a GST-pulldown experiment using purified p62-E1 protein produced in Expi293F cells. The results are similar to what was obtained using cell lysates and have been added into Extended Data Fig. 8c and the Extended Data Fig. 9d)

Extended Data Fig. 8c

Extended Data Fig. 9d

5. The sequence comparisons in extended data Fig. 8 are interesting and used to identify candidate sites of interaction (see also Fig. 6). However, these comparisons would be much stronger if the permissive CR 4 and CR5 domains were compared to the sequences of the non-permissive CR 1, 2 etc. In ED Fig 8, the comparison should include non-permissive host LDLR, and the mosquito homologue.

Response:

We aligned the 3D-structure of CR5 with two other resolved structures of CR domains, namely CR2 and CR6 which revealed two or three non-conservative amino acid differences in the sites we have been identified to be important for binding to GETV. See Extended Data Fig. 14b. Species that can definitely not be infected with GETV are not known and it is also unknown which receptor GETV uses in mosquitoes. Therefore, such a sequence comparison is not very useful.

6. Fig 2n presents the rather surprising finding that the deletion of MXRA8, which decreases GETV infection, can be rescued by expression of LDLR. What about the converse experiment--can the LDLR deletion or the LDLR/MXRA8 double deletion be rescued by MXRA8? The result currently is presented without any discussion of what it might mean.

Response:

We thank reviewers for suggestion. We have performed the experiment in the reverse way. As shown below, overexpression of MXRA8 can indeed partially rescue GETV infection in BHK-21 cells having LDLR deleted. The new data is added to Extended Data Fig. 7e-f. And we have added discussions around this topic in the revised manuscript (lines 196-201): "The ability of MXRA8 and LDLR to compensate the loss of each other in the infection of rGETV-EGFP in BHK-21 is intriguing, because MXRA8 is very different in structure from all the members of the LDLR family. This suggested that GETV have multiple ways of interacting with host cells and the infection *in vivo* might be dependent on actual expression of different receptors."

Extended Data Fig. 7

Minor points:

1. Please briefly describe the construction of the rGETV-eGFP reporter virus, which does not seem to be in ref. 33. Is this a cytoplasmic reporter? If the construct is similar to that of rGETV-mCherry, is there a problem with the cysteines in eGFP? Does the virus grow efficiently?

Response:

We have now added illustration of the construction of rGETV-EGFP into Extended Data Fig. 1a. Briefly, EGFP is expressed as individual proteins using viral subgenomic promoter and is, indeed, cytoplasmic. So there will not be any problem with the cysteines in EGFP. We have also performed the one-step growth curve experiments for all recombinant viruses (Extended Data Fig. 1d). All recombinant viruses are able to grow to high titers.

2. In the studies of the KO cells in extended data Fig. 4, why are there not results for both alleles of the various proteins? It would be quite surprising if both alleles had the same CRISPR-edited sequence for all of the KO. Please also explain if these are cell clones or populations.

Response:

Thanks for pointing this out for us. Just to make it clear, they are KO clones obtained by limiting dilution. We verified those clones by Western blot (Extended Data Fig. 6b) and sanger sequencing after PCR-amplifying the targeted genomic region and then subjecting the amplicon to T-A cloning. The edited sequence of the other allele was dropped by mistake. We have now amended the Extended Data Fig. 6a to include information on both alleles.

3. Throughout the paper, labeling of the constructs should accurately indicate the position of the tag. For example, the LDLR constructs state that they have N-terminal FLAG tags, but are designated LDLR-FLAG. This is confusing and not the usual convention.

Response:

Authors thank the reviewer for the suggestions and apologize for the confusion. As we have explained above, LDLR is actually expressed with C-terminal Flag. We have clarified this in the manuscript as well.

4. Line 66. Please adjust text to reflect that the mode of interaction (ie structural analysis) has also been determined for MXRA8 and LDLRAD3.

Response:

We thank reviewer for pointing this out. In the revised manuscript, we have changed the sentence to "In terms of mode of interaction, both LDLRAD3 and MXRA8 bind to the "canyon" between two protomers of the E spike on the surface of the alphaviruses, making simultaneous contact with

E1 and E2. On the other hand, VLDLR binds to the DIII domain of the SFV E1 protein close to the envelope membrane and does not interact with the E2 of SFV." Please see lines 69-73.

5. Line 71 is inaccurate. Both NCBI and Human protein atlas databases report very wide tissue expression of at least human VLDLR.

Response:

We thank reviewer for pointing this out. In the revised manuscript, we have changed the sentence to "Similarly, for SFV, ApoER2 is almost exclusively expressed in the central nervous system. Although VLDLR is expressed in multiple tissues, it has been observed that the absence of VLDLR or presence of antibodies against VLDLR do not completely block the infection of SFV in Vero, U2OS, A549, Huh7 and other cells". Please see lines 77-81.

6. Please adjust text in introduction to reflect that the Clark et al paper tested VLDLR from many species including mosquito.

Response:

We thank reviewer for pointing this out. In the revised manuscript, we have changed the sentence (lines 65-69) to " In addition, very low-density lipoprotein receptor (VLDLR) from many species (including mosquitoes) and apolipoprotein E receptor 2 (ApoER2) have recently been identified as viral receptors in vertebrate and invertebrate cells for SFV, EEEV and SINV."

7. It's very hard to see plaques in ED Fig 1e. Please fix figure.

Response:

We have replaced it with higher resolution image and due to other revisions it is now in Extended Data Fig. 1e.

8. Why does LDLR-FLAG show two bands in ED Fig 1g?

Response:

We have not analyzed the difference between those two bands of LDLR specifically. Because LDLR is N-glycosylated, our speculation is that the lower band is a form of LDLR with less degree of N-glycosylation, as suggested in literature (DOI: 10.1074/jbc.M113.545053).

Reviewer #3 (Remarks to the Author)

In this study, the authors identified LDLR as a crucial receptor involved in entry of several viruses belonging to the Alphavirus genus into target cells, whereby Getah virus was the main focus of the study due to biological safety reasons. The overall experimental strategy was well-designed. However, there is major concern if LDLR is indeed the crucial receptor for these viruses without validation experiments with in-vivo knockout model of LDLR. In-vivo model of LDLR knockout are well established and has been utilized for diseases such as hypercholesterolemia.

Major points:

1) The authors have constructed an expression plasmid library for 150 mouse membrane proteins, in order to identify the specific protein(s) which positively affect alphavirus infections. What are the selection criteria of these proteins, have these proteins been shown in the past to be associated with the infection capacity of other viruses?

Response:

We acknowledge that the library used in this study is of limited size. It was obtained as part of our effort to build a library of all the murine membrane proteins (about 10,026 proteins in total; the completion of this project may take a decade). By the time this study was initiated we had validated the expression constructs for the first 150 membrane proteins (Extended Data Table 1). Initially, we did not know whether or not they are related to the virus infection. Aside of our main findings, this study represents a proof that construction and use of such libraries is a valid approach. We believe that in our larger library, which is still under construction, we will find more and more unknown host factors that can affect the alphavirus life cycle. However, since our identification of LDLR from this limited size library is kind of serendipity, we decided to move this result to Extended Data Fig. 2b to avoid misleading the audience.

2) It will be useful for the authors to provide a list of the tested proteins including those which were negative hits in this study, as it can serve as an important source of information for future works by the readers.

Response:

A list of the 150 membrane proteins is now presented as the Extended Data Table 1.

3) Figure 2j: Can the authors provide a clear description on the y-axis of the graph? Is the y-axis presented as relative ratio of internalization:binding data? The error bars are too huge in my opinion to deduce that there is an effect on the virus internalization, while there are some data points which showed similar levels of vRNA as the WT.

Response:

Thank you for the comment. We apologize for the confusion. Our initial intention was to normalize the amount of internalized virions to the amount of bound. We have now realized that this is unnecessary as is shown by literatures (DOI: 10.1038/s41586-018-0121-3;). We have therefore removed this ratio and combined the data of the bound and internalized in the same way as in the literatures (Fig. 2a). We also went back to check the reason for the unusually large error bars. The consistency of the qPCR is significantly improved after we switch to a different qPCR kit. The new data is packed into Fig. 2a in the revised manuscript.

Fig. 2a

4) In Figure 2t, the authors showed that JEV is among the virus which are not affected by LDLR deficiency in the host cells. However, this is in contrast to a previous finding (<https://www.ncbi.nlm.nih.gov/pmc/articles/PMC8238074/>) which mechanistically demonstrated that LDLR is an important host factor required for JEV entry. The authors will need to address this contrasting data.

Response:

We thank reviewer for pointing this out. This is a very interesting finding. The apparent contradiction could be the result of different cell lines used in the study. We used BHK-21 cell, which is a baby hamster kidney fibroblast cell, while the literature referred by reviewer used A549, a human lung cancer cell line. It is possible that the dependency of JEV on LDLR as receptor is different in these two cell lines. JEV might have an alternative, highly expressed receptor in BHK-21 cells, making it insensitive the deletion of LDLR. Since JEV is not in the focus of this study, we did not perform further experiments to test our hypothesis and removed results with JEV from the manuscript.

5) Although the mouse sera used in this study was collected from mice subcutaneously injected with GST-LBD, how did the author confirm the presence of anti-LDLR antibodies in the sera generated from their mouse model? Was there any neutralization test being performed using the generated sera? The authors need to convincingly show that the inhibition observed on the rGETV-EGFP infection within the different cell lines tested are indeed due to the presence of anti-LDLR antibodies.

Response:

Thanks for your suggestion. We have now included ELISA assay of the anti-LDLR mouse sera against purified GST-LBD in Extended Data Table 2. Further, we performed antibody validation and cross-reactivity experiments. As shown in the figure below, the LDLR antibody we used for neutralization studies could react with GST-LBD and full-length LDLR-Flag expressed in HEK 293T cells, but not with VLDLR-Flag. The result indicated that our antibody could specifically recognize LDLR but not its highly related family member VLDLR. We included the new data in Extended Data Fig. 11.

Extended Data Fig. 11

6) It is not convincing with the rather low levels of infection efficiency by the different viruses generated by the authors in Figure 1, it is perplexing with the extremely low levels of EGFP⁺ cells illustrated in Figures 3d, 6b and 6g, within the WT or empty vector control groups, which are rather contrasting and not even close to the initial data in Figure 1d showed an approximately 10% infection by GETV. Another concern from this point is that the rather low infection levels of the WT group might result in a biased over-interpretation of the efficiency of LDLR expression in the rest of the experiments in this study.

Response:

The differences in infection efficiency between figures are due to the different systems used for the study. In Fig. 1 we used lentivirus pseudotypes and in Figures 3d, 6b and 6g we used authentic viruses. Pseudo-viruses and real viruses have different infection efficiencies. In addition, the methods of preparation and quantification of virus titers are different between pseudo-viruses and authentic viruses. Hence the different infection rates are not unexpected between them.

7) Validation experiments of virus infectivity and pathogenesis with in-vivo knockout model of LDLR is essential.

Response:

Thanks for the suggestion. We agree that *in vivo* studies in a LDLR KO mouse would be helpful for evaluating the importance of LDLR in the GETV-caused pathology. However, we think the focus of our current study is the discovery of LDLR as a GETV receptor. In addition, the *in vivo* experiments require LDLR-KO suckling mice, which we do not currently have. Therefore, we have to leave this to our next manuscript.

8) Figure 6b and c: It is inconclusive for the authors to deduce that the CR4 region are also essential for promoting GETV infection as there were still statistically significant upregulation of the EGFP⁺ cells and viral titers in the absence of CR4. Line 223: Was the promoting effect in the absence of CR4 statistically compared to CR5? It is inaccurate for the authors to state that "deletion of CR4 statistically reduced the promoting effect of LDLR expression", as the statistics performed were on the upregulation of EGFP⁺ cells and viral titers relative to the WT group.

Response:

Thank you for pointing this for us. Indeed, this is due to lack of statistics analysis in the figure. We have now amended Fig. 6b and 6c with statistics analysis of the negative impact of EGFP+ cells and viral titers of CR4 and CR5 mutants relative to the full-length LDLR. We further modified the text (lines 285-290) as follows: “After deletion of CR4, EGFP+ cells and viral titers were still significantly upregulated compared with those in the group without overexpression of LDLR protein, but to a much lesser extent compared with those overexpressing full-length LDLR protein (Fig. 6b-d). This suggests that both CR4 and CR5 are essential for the infection of GETV, with CR5 playing the most dominant role”.

Minor comments:

1) Figure 1b: The authors should add another sub-section on the lower end of y-axis, as the current graphical illustration gives a rather confusing perspective that the vector and LDLRAD3 groups do not generate any viral titer whereas the actual scenario is that the vector has a 100% relative viral RNA level.

Response:

We thank the reviewer for the suggestion. Fig. 1b is now modified according to the suggestion.

2) Figure legend for Figure 6: PDB IDs have to be in uppercase.

Response:

We thank reviewer for the suggestion. We have modified the legend for Fig. 6 accordingly.

3) There are a few typing errors across the manuscript

Response:

Thank you for your comment. We have tried to remove those typos.

REVIEWER COMMENTS

Reviewer #1 (Remarks to the Author):

In revising the manuscript, the authors have strengthened it by now including data with chimeric Sindbis viruses for Semliki Forest virus (SFV), Ross River virus (RRV), and Bebaru virus (BEBV). Because they are Sindbis virus chimeras, these particles should have properly processed and assembled spike proteins. The authors show that binding and internalization are enhanced by LDLR overexpression. Additional data are provided to support that the LDLR LBD can precipitate from solution the spike proteins of SFV, BEBV, and RRV. Although the RRV band on the western blot is weak, given the other supporting data provided in the manuscript, this is less of a concern.

The observation that VLDLR and ApoER2 (LRP8) are also probably receptors for GETV is not surprising given the known role of these LDLR-related molecules as receptors for multiple alphaviruses, including SFV, which is another Old World alphavirus that is specifically studied here.

While this reviewer understands that the authors would want to carry additional work to further support this finding, the discussion section of the manuscript should address that VLDLR and ApoER2 were not on the list of genes tested, and that given the homology of LDLR with these proteins and known roles of VLDLR and ApoER2 as alphavirus receptors, teasing out their potential roles as GETV receptors would be an important area of future investigation.

In terms of the BLI assays to demonstrate an interaction of the GETV spike protein with VLPs, the authors are correct in stating that an approach in which avidity is used to capture the VLPs is a standard approach in the field. My concern was that the "KD" being reported was not a valid assessment of affinity. For BLI data to be robust, usually, multiple concentrations are run in the experiment, and both the experimental data and fit are provided in graph form. Extended Data Fig. 8d shows only a single concentration and no fit, and the signal is very weak (0.12 response units). Such a weak signal, although seemingly specific given the results with the negative control, at best suggests weak binding. Unless how the data were fit can be shown, I would suggest the authors consider including in the paper both the VLP and p62-E1 binding data to demonstrate that there is a specific, detectable interaction but not attempt to report a KD.

Otherwise, the authors have adequately answered all of my comments and concerns.

Reviewer #2 (Remarks to the Author):

This revised manuscript by Zhai et al. has addressed most of the numerous issues raised by the 3 reviewers. The revisions and additions have considerably strengthened and clarified the paper, and overall the paper now makes a strong case for the role of LDLR as a receptor for Getah Virus, and also for SFV, RRV, and BEBV. Some gaps in the data and other issues still need to be addressed.

Major points:

1. The authors now use recombinant reporter viruses for SFV, RRV, and SINV in addition to their lentivirus pseudotypes. However, it is important to note that all of these have mCherry linked to the N-terminus of the E2 protein. While this can be a useful construction for live cell imaging etc, it can definitely modify and/or inhibit the binding of the virus to the receptor used by authentic virus. It is thus unfortunate that these are the reporter viruses that are used to confirm receptor binding in this study, although this is mitigated by the inclusion of the p62-E1 binding data presented later. The authors should in addition confirm their results with authentic virus. Infectivity results would be adequate, or the binding assays shown in Extended data Fig. 6, which don't require

mCherry-labeled virus. Please also add a comment around lines 135-138 to point out that authentic GETV (w/o mCherry on E2) binds to LDLR (the data are very strong on this point).

2. It is critical to confirm that the various LDLR mutants are expressed on the cell surface. The mutants were C-terminally tagged, and the new confocal microscopy data support their delivery to the cell surface. This is not a quantitative assay but looks convincing. The flow cytometry on the permeabilized cells does not provide evidence for cell surface delivery, and the presentation of the cytometry should be revised to simply say that the steady state levels of the various mutants are approximately the same.
3. This reviewer disagrees with the response to the request to test the role of VLDLR and ApoER2 in GETV infection. The data shown in the rebuttal demonstrate that both proteins can act as receptors for GETV (although it would be important to perform this experiment with GETV without the mCherry-E2 tag, see point 1). The authors were not being asked to test all representative alphaviruses, but simply to test GETV. The authors should add this data, using GETV without the mCherry-E2 tag (unclear from the rebuttal which virus was used).
4. The comparison of the sequence of LDLR CR 4 and CR5 domains in the extended data should include the mosquito sequence. While mosquito LDLR has not yet been tested as a receptor, it would be quite helpful to know if the sequence suggests that it could serve as a receptor in the mosquito. This is a simple comparison to do and the authors should add it.
5. It appears that the purified GETV p62-E1 protein retained the E1 and E2 transmembrane domains, but the purified protein was used for biochemical studies in the absence of detergent. Please clarify if this is correct and if so how the solubility of the protein was maintained in PBS. A diagram of the construct would be helpful, as it appears to be different than the p62-E1 hybrid protein used for example in the studies of Voss et al Nature 2010. Also, were the p62-E1 protein constructs of the other viruses prepared similarly? This should be added to the methods.

Minor points:

1. SINV was recently shown to bind avian MXRA8 (DOI: 10.1016/j.cell.2023.09.007), so it is important not to overstate the "global" role of LDLR-family members as alphavirus receptors.
2. It would be helpful to include a comment in the methods explaining why the commercial antibody vs. their generated antibody is used for specific experiments.
3. The symbols in graph Fig 2B are very hard to tell apart; the authors should consider changing colors or making them more distinct.

Reviewer #3 (Remarks to the Author):

The authors have addressed my comments with the necessary experimental data/justification.

REVIEWER COMMENTS

Reviewer #1 (Remarks to the Author):

In revising the manuscript, the authors have strengthened it by now including data with chimeric Sindbis viruses for Semliki Forest virus (SFV), Ross River virus (RRV), and Bebaru virus (BEBV). Because they are Sindbis virus chimeras, these particles should have properly processed and assembled spike proteins. The authors show that binding and internalization are enhanced by LDLR overexpression. Additional data are provided to support that the LDLR LBD can precipitate from solution the spike proteins of SFV, BEBV, and RRV. Although the RRV band on the western blot is weak, given the other supporting data provided in the manuscript, this is less of a concern.

The observation that VLDLR and ApoER2 (LRP8) are also probably receptors for GETV is not surprising given the known role of these LDLR-related molecules as receptors for multiple alphaviruses, include SFV, which is another Old World alphavirus that is specifically studied here.

While this reviewer understands that the authors would want to carry additional work to further support this finding, the discussion section of the manuscript should address that VLDLR and ApoER2 were not on the list of genes tested, and that given the homology of LDLR with these proteins and known roles of VLDLR and ApoER2 as alphavirus receptors, teasing out their potential roles as GETV receptors would be an important area of future investigation.

Response:

Authors thank you for the suggestion. We have now included related results of VLDLR and ApoER2 into Extended Data Fig. 13 and inserted the following discussion into the revised manuscript (lines 337-344):

“Because SFV uses VLDLR and ApoER2 as receptors, which were not on the list of our limited library of membrane proteins yet also belong to LDLR family, we tested whether these two receptors could mediate cell entry of GETV. Not surprisingly, both VLDLR and ApoER2 significantly promoted the infectivity of authentic GETV in BHK-21 (Extended Data Fig. 13). The association of the infectivity of certain alphavirus with LDLR family proteins is intriguing but requires more systematic analysis in the future to confirm this as a general rule.”

In terms of the BLI assays to demonstrate an interaction of the GETV spike protein with VLPs, the authors are correct in stating that an approach in which avidity is used to capture the VLPs is a standard approach in the field. My concern was that the "KD" being reported was not a valid assessment of affinity. For BLI data to be robust, usually, multiple concentrations are run in the experiment, and both the experiment data and fit are provided in graph form. Extended Data Fig. 8d shows only a single concentration and no fit, and the signal is very weak (0.12 responses units). Such a weak signal, although seemingly specific given the result with negative control, at best suggests weak binding. Unless how the data were fit can be shown, I would suggest the authors consider including in the paper both the VLP and p62-E1 binding data to demonstrate that there is a specific, detectable interaction but not attempt to report a kD.

Response:

Thanks for pointing this out for us. We have now provided both the binding data of LDLR LBD to GETV VLPs and purified p62-E1 protein, we also removed "KD" from the graph. As shown in the Extended Data Fig. 9f and 9g, the analysis revealed that both the GETV p62-E1-Strep and GETV

VLPs display a specific binding to GST-LBD but not to GST.

Otherwise, the authors have adequately answered all of my comments and concerns.

Reviewer #2 (Remarks to the Author):

This revised manuscript by Zhai et al. has addressed most of the numerous issues raised by the 3 reviewers. The revisions and additions have considerably strengthened and clarified the paper, and overall the paper now makes a strong case for the role of LDLR as a receptor for Getah Virus, and also for SFV, RRV, and BEBV. Some gaps in the data and other issues still need to be addressed.

Major points:

1. The authors now use recombinant reporter viruses for SFV, RRV, and SINV in addition to their lentivirus pseudotypes. However, it is important to note that all of these have mCherry linked to the N-terminus of the E2 protein. While this can be a useful construction for live cell imaging etc, it can definitely modify and/or inhibit the binding of the virus to the receptor used by authentic virus. It is thus unfortunate that these are the reporter viruses that are used to confirm receptor binding in this study, although this is mitigated by the inclusion of the p62-E1 binding data presented later. The authors should in addition confirm their results with authentic virus. Infectivity results would be adequate, or the binding assays shown in Extended data Fig. 6, which don't require mCherry-labeled virus. Please also add a comment around lines 135-138 to point out that authentic GETV (without mCherry on E2) binds to LDLR (the data are very strong on this point).

Response:

To address the reviewer's concern, we performed new binding experiments using authentic viruses without mCherry reporter. The results are consistent with those using recombinant reporter viruses and are added in Extended Data Fig. 6d and 6e.

Fig. 1a and 1b are indeed results from rGETV-GFP without mCherry on E2, which was noted in the figure legend but not in the main text. We have now moved the introduction to rGETV-GFP to lines 136-139, as the following: "To rule out a potential interference of mCherry on E2 to the results, we further generated another recombinant GETV reporter virus (rGETV-EGFP) for validation, which had EGFP expressed as an individual protein under the control of the viral subgenomic promoter (Extended Data Fig. 1)."

As a separate note, we also have data for authentic GETV without any reporter genes, which are displayed in Fig. 2a and Extended Data Fig. 5.

2. It is critical to confirm that the various LDLR mutants are expressed on the cell surface. The mutants were C-terminally tagged, and the new confocal microscopy data support their delivery to the cell surface. This is not a quantitative assay but looks convincing. The flow cytometry on the permeabilized cells does not provide evidence for cell surface delivery, and the presentation of the cytometry should be revised to simply say that the steady state levels of the various mutants are approximately the same.

Response:

Thanks for pointing this out. We have modified the text as follows: "which have similar steady state expression levels (Extended Data Fig. 4a) and membrane localization (Extended Data Fig. 4b)." (lines 285-287); "Again, it was observed that mutant proteins had approximately same steady state expression levels (Extended Data Fig. 4a) and had maintained their membrane localization (Extended Data Fig. 4b)." (lines 310-313).

3. This reviewer disagrees with the response to the request to test the role of VLDLR and ApoER2

in GETV infection. The data shown in the rebuttal demonstrate that both proteins can act as receptors for GETV (although it would be important to perform this experiment with GETV without the mCherry-E2 tag, see point 1). The authors were not being asked to test all representative alphaviruses, but simply to test GETV. The authors should add this data, using GETV without the mCherry-E2 tag (unclear from the rebuttal which virus was used).

Response:

Authors thank the reviewer for the comment and suggestion. The results of VLDLR and ApoER2 shown in previously in the rebuttal letter were indeed obtained using authentic GETV-HN strain without any tag. We have now included these results into Extended Data Fig. 13 and inserted the following discussion into the revised manuscript (lines 337-344): “Because SFV uses VLDLR and ApoER2 as receptors, which were not on the list of our limited library of membrane proteins yet also belong to LDLR family, we tested whether these two receptors could mediate cell entry of GETV. Not surprisingly, both VLDLR and ApoER2 significantly promoted the infectivity of authentic GETV in BHK-21 (Extended Data Fig. 13). The association of the infectivity of certain alphavirus with LDLR family proteins is intriguing but requires more systematic analysis in the future to confirm this as a general rule.”

4. The comparison of the sequence of LDLR CR 4 and CR5 domains in the extended data should include the mosquito sequence. While mosquito LDLR has not yet been tested as a receptor, it would be quite helpful to know if the sequence suggests that it could serve as a receptor in the mosquito. This is a simple comparison to do and the authors should add it.

Response:

Thanks for your suggestion. We searched the NCBI-database and found a lipophorin receptor from *Aedes aegypti*, which has 36% amino acid sequence identity to human LDLR. We have now added the sequence of CR4 and CR5 of this to the Extended Data Fig. 15a, which shows a conservation of the three important sites in CR4 and four out of five in CR5 of the lipophorin receptor. Interestingly, all residues revealed to be crucial for GETV-LDLR interaction are identical in the mosquito protein.

5. It appears that the purified GETV p62-E1 protein retained the E1 and E2 transmembrane domains, but the purified protein was used for biochemical studies in the absence of detergent. Please clarify if this is correct and if so how the solubility of the protein was maintained in PBS. A diagram of the construct would be helpful, as it appears to be different than the p62-E1 hybrid protein used for example in the studies of Voss et al Nature 2010. Also, were the p62-E1 protein constructs of the other viruses prepared similarly? This should be added to the methods

Response:

Authors thank the reviewer for the comment and apologize for lack of description here. For BLI binding assay, we obtained soluble GETV p62-E1 protein by deleting transmembrane domains of E1 and E2. Thus, the purified soluble GETV p62-E1 protein use in this study was similar to that described by Voss et al (Nature 2010). We have now included this information into Materials and Method section (lines 652-657). Additionally, we have added a diagram of the recombinant protein design in Extended Data Fig. 8c and, in order to avoid confusion, we renamed this purified soluble protein to GETV p62-E1-Strep.

For co-precipitation assay, we used a different design. Expression plasmids of p62-E1-HA protein were constructed for GETV, SFV, RRV and BEBV. In these constructs transmembrane domains were retained. For the detailed methods, see lines 597-601. We also added diagrams of these constructs in Extended Data Fig. 9a.

Minor points:

1. SINV was recently shown to bind avian MXRA8 (DOI: 10.1016/j.cell.2023.09.007), so it is important not to overstate the "global" role of LDLR-family members as alphavirus receptors.

Response:

We thank reviewers for the suggestion. In the revised manuscript, we have changed the sentence (lines 106-108) to “It remains unclear whether these receptors are used differently in different cell types, or whether other proteins function as receptors for various alphaviruses”.

In addition, we have included the new reference (also describing avian MXRA8 as receptors for WEEV) into manuscript. Please see reference 19.

2. It would be helpful to include a comment in the methods explaining why the commercial antibody vs. their generated antibody is used for specific experiments

Response:

We have added the following note to the materials and methods section (lines 707-709): “In some western blot and flow cytometry experiments, commercial antibodies were used because these experiments were performed before our own anti-LDLR antibodies become available”.

3. The symbols in graph Fig. 2B are very hard to tell apart; the authors should consider changing colors or making them more distinct.

Response:

Thanks for pointing this out. We have changed the colors use in graph Fig. 2b.

REVIEWERS' COMMENTS

Reviewer #2 (Remarks to the Author):

The authors have answered all of the points raised by the reviewers. Nice job. I have no further comments.